# Rethinking KV Cache Eviction via a Unified Information-Theoretic Objective

**Jiaming Yang** [1 2]  **Chenwei Tang** [1 2]  **Liangli Zhen** [3]  **Jiancheng Lv** [1 2]

## Abstract

Key–Value (KV) caching is essential for large language model inference, yet its memory overhead poses a critical bottleneck for long-context generation. Existing eviction policies predominantly rely on empirical heuristics, lacking a rigorous theoretical foundation. This work rethinks KV cache eviction through the lens of the *Information Bottleneck* principle. Under a linear–Gaussian surrogate of attention, we derive a closed-form mutual information objective that characterizes the effective information capacity of a retained KV cache subset. This formulation reveals that a wide range of existing eviction strategies can be interpreted as different approximations of the same capacity-maximization principle. Guided by this insight, we introduce CAPKV, a capacity-aware eviction method that directly targets information preservation via a log-determinant approximation using statistical leverage scores. This approach replaces heuristic selection with a theoretically grounded mechanism that preserves the maximum predictive signal. Extensive experiments across multiple models and long-context benchmarks show that CAPKV consistently outperforms prior methods, achieving a better trade-off between memory efficiency and generational fidelity. Our code is available in this link.

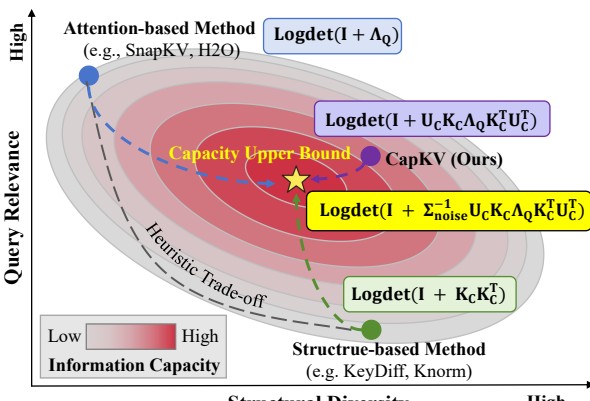

*Figure 1.* **A unified information-theoretic view of KV cache eviction.** The horizontal axis reflects structural diversity of retained KV entries, while the vertical axis reflects query relevance, measuring alignment with likely future queries. KV cache eviction can be viewed as maximizing an information-capacity objective, with existing methods corresponding to different approximations.

## 1. Introduction

In Large Language Model (LLM) inference, Key-Value (KV) caching is a critical optimization that avoids recomputation at each decoding step and significantly reduces in-

ference latency by storing the key and value representations of previously generated tokens (Pope et al., 2022; Shi et al., 2024). This mechanism is essential for accelerating inference and enabling real-time deployment in latency-sensitive applications, such as on-device agents and embodied systems (Liu et al., 2025a;b). However, the linear growth of the KV cache with context length creates a formidable memory bottleneck, particularly in long-context scenarios (Xu et al., 2024; Recasens et al., 2025). While the inherent robustness of Transformer attention to input perturbations facilitates significant redundancy in the KV cache during autoregressive decoding (Li et al., 2024c; Ribar et al., 2024), existing KV cache compression strategies primarily focus on two dimensions: representation compression, which utilizes low-precision quantization or low-rank approximation (Hooper et al., 2024; Liu et al., 2024b), and structural compression, which dynamically prunes or merges cache entries (Zhang et al., 2024; Liu et al., 2024a). Among these, importance-based cache eviction has emerged as a particularly promising structural approach due to its model-agnostic nature and seamless integration into inference pipelines. However, a fundamental challenge remains: *designing a low-overhead scoring mechanism that precisely identifies and prunes low-contribution entries without sacrificing the model's predictive performance or generational fidelity* (Li et al., 2024a).

Among structural approaches, importance-based KV cache

[1]College of Computer Science, Sichuan University, Chengdu, 610065, China [2]Engineering Research Center of Machine Learning and Industry Intelligence, Ministry of Education, Chengdu, 610065, China [3]Institute of High Performance Computing, Agency for Science, Technology and Research (A*STAR), Singapore 138632, Singapore. Correspondence to: Chenwei Tang <tangchenwei@scu.edu.cn>.

*Proceedings of the 43$^{rd}$ International Conference on Machine Learning*, Seoul, South Korea. PMLR 306, 2026. Copyright 2026 by the author(s).

eviction is particularly promising due to its model-agnostic nature. Existing approaches generally follow two primary paradigms: attention-pattern-based eviction, which removes tokens with low historical attention scores (Li et al., 2024b; Zhang et al., 2023), and structure-aware eviction, which prioritizes representational diversity based on intrinsic geometric properties of the KV cache (Park et al., 2025; Devoto et al., 2025). While empirically effective, both paradigms are largely driven by heuristics and lack a unified, interpretable optimization objective, leaving the underlying mechanisms of information preservation poorly understood. To bridge this gap, we rethink KV cache eviction from an information-theoretic perspective. By modeling the KV cache as a collection of linear communication channels (Schwartz et al., 1995), we introduce a unified analytical framework grounded in the *Information Bottleneck* principle (Tishby et al., 2000; Tishby & Zaslavsky, 2015). As illustrated in Figure 1, we demonstrate that diverse existing strategies, whether attention-pattern-based eviction or structure-aware eviction, can be interpreted as implicit approximations of a single, global information-preservation objective: *maximizing the mutual information between latent future queries and transmitted representations*.

Specifically, we adopt a linear-Gaussian abstraction of attention dynamics as a tractable tool to enable rigorous analysis. Under this formulation, attention computation is interpreted as a linear transmission process, reducing the eviction problem to an optimization task: *selecting a subset of B channels that maximizes the mutual information under a fixed cache budget*. Crucially, we demonstrate that the resulting channel capacity metric is strongly correlated with downstream task performance. This perspective not only unifies existing heuristic strategies as disparate approximations of the same objective, i.e., the objective in the yellow block in Figure 1, but also provides clear design principles for developing new eviction mechanisms. Building on these insights, we propose CAPKV, a principled eviction strategy that explicitly optimizes the cache's information capacity. Specifically, CAPKV bypasses empirical scoring rules by greedily selecting tokens with the highest statistical leverage scores derived from a simplified capacity matrix, providing a deterministic approximation, i.e., the objective in the purple block in Figure 1, to the optimal information-preservation objective. The main contributions of this work are:

- We establish a principled analytical framework grounded in the *Information Bottleneck* principle, formulating KV cache eviction as mutual information maximization within a linear-Gaussian channel to unify disparate heuristic strategies under a single fundamental objective.

- We propose CAPKV, a novel cache eviction method that explicitly optimizes cache information capacity.

By employing statistical leverage scores derived from a simplified capacity matrix, CAPKV provides a deterministic and theoretically grounded alternative to heuristic scoring rules.

- Extensive experiments across multiple model architectures and long-context benchmarks demonstrate that CAPKV consistently outperforms prior state-of-the-art methods, achieving a superior trade-off between generational fidelity and memory efficiency.

## 2. Related Work

Existing KV cache eviction methods primarily focus on identifying historical tokens that are most informative for future generations, following two main trajectories. The first leverages attention-based signals as proxies for token importance. For example, SnapKV (Li et al., 2024b) identifies key positions via local-window attention distributions, while Expected Attention (EA) predicts future query relevance to bypass the need for exact attention matrices (Devoto et al., 2025). The second trajectory exploits the intrinsic structural properties of the KV cache to minimize redundancy. For instance, KeyDiff (Park et al., 2025) employs pairwise vector differences to ensure representation uniqueness, and Knorm (Devoto et al., 2024) utilizes $\ell_2$-norms as lightweight importance metrics. More recently, methods based on output perturbation (Feng et al., 2025) have attempted to quantify token sensitivity. However, these approaches remain largely driven by local heuristics or instance-level criteria. They often treat tokens in isolation, failing to capture the complex redundancy and complementarity across cache entries. Consequently, they lack a holistic characterization of the collective information content, necessitating a more principled framework to evaluate the retained set's global utility.

The information bottleneck principle (Tishby et al., 2000; Chechik et al., 2003) provides a rigorous foundation for balancing information compression against task relevance under capacity constraints. In the broader context of representation learning, the information bottleneck framework is frequently employed to analyze how intermediate layers filter irrelevant noise while preserving task-critical signals, often through variational or linear-Gaussian formulations (Tishby & Zaslavsky, 2015; Wieczorek & Roth, 2020). Unlike heuristic eviction rules that rely on local proxies, information bottleneck offers a global perspective by characterizing how a collection of representations collectively maintains relevant information. This information-theoretic lens is uniquely suited for reasoning about capacity-limited selection, as it treats the KV cache not as a list of independent scores, but as a communication channel with a finite information volume. This perspective directly motivates our formulation of KV cache eviction as an explicit mutual information maximization problem.

# 3. Methodology

## 3.1. Problem Definition

Modern LLMs predominantly adopt an autoregressive Transformer decoder architecture, in which self-attention constitutes the core computation (Deng et al., 2025). For simplicity, we describe the single-head self-attention mechanism. At any decoding step $t$, let $h_t$ be the hidden state, which is projected into query $q_t = W_Q h_t$, key $k_i = W_K h_i$, and value $v_i = W_V h_i$ vectors, where $i \leq t$. The attention output $\text{Attn}(q_t)$ is computed by aggregating values weighted by their relevance to the query:

$$\text{Attn}(q_t) = W_O \sum_{i \leq t} a_{ti} v_i, \tag{1}$$

where $a_{ti} = \text{softmax}(q_t^\top k_i / \sqrt{d})$ denotes the attention score assigned to token $i$.

To avoid recomputing these representations during autoregressive decoding, the set of all historical keys $K = [k_1, \ldots, k_t]$ and values $V = [v_1, \ldots, v_t]$ is stored in GPU memory as the *KV cache*. However, the linear scaling of this cache with context length necessitates structural compression to remain within hardware memory limits. Formally, given a fixed memory budget $B$, an eviction policy must select an optimal index set $C \subset \{1, \ldots, t\}$ with $|C| \leq B$. The resulting compressed cache, $Z_C = \{(k_i, v_i)\}_{i \in C}$, is then utilized for subsequent computations. The goal of KV cache eviction is to find the subset $C$ that minimizes information loss, ensuring that the compressed representation $Z_C$ preserves the maximum predictive signal required for accurate auto-regressive generation.

## 3.2. Rethinking in Information Bottleneck Perspective

The information bottleneck principle provides a theoretical framework for analyzing representation compression under capacity constraints (Tishby & Zaslavsky, 2015; Siddiky et al., 2024). In its classical formulation, information bottleneck seeks an intermediate representation $T$ that minimizes the information extracted from an input $X$ while maximizing the predictive information for a target $Y$:

$$\min_T I(X;T) - \beta I(T;Y), \tag{2}$$

where $I$ measures the mutual information between two variables, and $\beta$ controls the trade-off between compression and predictive information.

While KV cache eviction is fundamentally a form of representation compression, it diverges from the classical information bottleneck setting. In our scenario, queries, keys, and values are intermediate activations produced by a fixed, pretrained model during inference. There is no learned encoder or explicit optimization over $T$, instead, compression

is directly enforced through a hard cache capacity constraint $|C| \leq B$. Consequently, the compression term $I(X;T)$ in Eq. (2) is not an explicit optimization objective. And the optimization objective shifts from determining ***how much*** to compress to identifying ***which*** representations to preserve to maximize the utility of the limited capacity.

A key insight is that *the primary source of uncertainty in autoregressive decoding resides in future queries*. While historical keys and values are fixed once generated, future queries depend on unobserved tokens and evolving trajectories. Therefore, the goal of an eviction policy is to select a subset $Z_C$ that can robustly support the diversity of these latent future queries. Motivated by this, we model the future query $q$ as a random variable and the retained KV subset $Z_C$ as a fixed conditioning set. Let $Y$ denote the attention output derived from $Z_C$. We quantify the information preservation of the cache through the conditional mutual information:

$$\mathcal{L}_C = I(q; Y \mid Z_C), \tag{3}$$

which measures the information content about the query $q$ that is successfully transmitted through the bottleneck of the selected KV pairs to the output $Y$. Maximizing $\mathcal{L}_C$ encourages the retained cache entries to support a broad range of possible future queries, while discarding representations with limited predictive contribution. Although this formulation does not optimize the full information bottleneck functional in Eq. (2), it directly captures the predictive information term, with the cache capacity constraint implicitly enforcing compression.

## 3.3. Linear–Gaussian Surrogate Transmission Model

To obtain a tractable approximation to the objective $\mathcal{L}_C$ in Eq. (3), we introduce a surrogate transmission model that interprets the attention computation induced by a retained KV subset as an information channel mapping future queries to model outputs (Forney & Ungerboeck, 2002). This surrogate is adopted solely for analytical tractability and is not intended to faithfully model the full nonlinear dynamics of Transformer architectures.

**Assumption 3.1** (*linear-Gaussian surrogate transmission model*). *For analytical purposes, we assume that, conditioned on a retained KV subset $Z_C = \{(k_i, v_i)\}_{i \in C}$, the attention-induced output $Y$ can be approximated by a linear-Gaussian model of the form:*

$$Y = U_C K_C q + \varepsilon, \tag{4}$$

*where $K_C \in \mathbb{R}^{|C| \times d}$ stacks the retained keys, $U_C \in \mathbb{R}^{m \times |C|}$ stacks the corresponding value-induced output directions (e.g., $u_i = W_O v_i$), and $\varepsilon$ aggregates unmodeled effects, including softmax nonlinearities, interactions with other attention heads, and downstream transformations.*

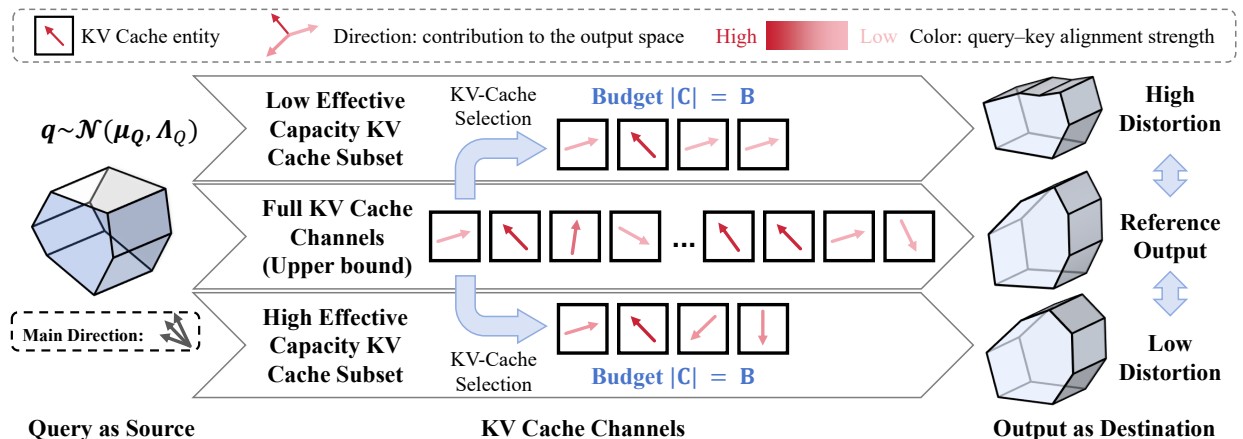

*Figure 2.* **Information-Theoretic Perspective of KV Cache Eviction**. We model cached key–value pairs as communication channels mapping future queries to attention outputs. **Query as Source**: Future queries are characterized as stochastic signals sampled from $q \sim \mathcal{N}(\mu_Q, \Lambda_Q)$. **KV Cache Channels**: Each cached entities induces an effective channel, the vector direction signifies its contribution to the output space, while color intensity encodes query–key alignment. Given a fixed budget $|C| = B$, diverse channel orientations maximize the information capacity (Theorem 3.2), whereas redundant directions lead to capacity loss. The full KV cache serves as the theoretical upper bound. **Output as Destination**: Output fidelity is evaluated against the full-cache reference, demonstrating that higher effective capacity directly correlates with lower representational distortion.

Consistent with the information bottleneck perspective, we treat the future query $q$ as a stochastic signal following a Gaussian distribution, $q \sim \mathcal{N}(\mu_Q, \Lambda_Q)$. We further assume the noise $\varepsilon \sim \mathcal{N}(0, \Sigma_{\text{noise}})$ is independent of $q$. Under Assumption 3.1, Eq. (4) defines a *linear-Gaussian surrogate transmission channel* from the future query $q$ to the attention output $Y$, conditioned on the retained KV subset $Z_C$. In this framework, the KV-cache entries are no longer viewed merely as stored activations, but as the structural parameters of a communication channel. The eviction problem thus reduces to a channel design problem: selecting a subset of entries $C$ that maximizes the information capacity of this surrogate channel, thereby ensuring the most efficient preservation of context-relevant signals.

### 3.4. Unified Information Objective

Under Assumption 3.1, the conditional distribution $P(Y|q, Z_C)$ and the marginal distribution $P(Y|Z_C)$ are Gaussian. While the mean $\mu_Q$ affects the conditional mean of $Y$, it does not influence the mutual information between $q$ and $Y$. We can characterize information preservation in closed form by viewing Eq. (4) as a linear-Gaussian transmission channel (see complete derivation in Appendix A).

**Theorem 3.2.** *Under Assumption 3.1, the conditional mutual information between the query $q$ and the output $Y$ admits the following closed form:*

$$I(q; Y \mid Z_C) =$$
$$\frac{1}{2} \log \det \left( I + \Sigma_{\text{noise}}^{-1} U_C K_C \Lambda_Q K_C^\top U_C^\top \right). \tag{5}$$

Under Assumption 3.1, the output distribution is Gaussian

both conditionally and marginally, and the mutual information reduces to the difference of two Gaussian entropies: $H(Y \mid Z_C) - H(Y \mid q, Z_C)$. This difference depends only on the corresponding covariance matrices and is invariant to the query mean. Applying standard determinant identities yields the log-determinant form. As shown in Figure 2, Theorem 3.2 characterizes the information capacity of the surrogate transmission channel induced by the retained KV subset. The expression highlights four key factors governing information preservation: i) **Key Accessibility** ($K_C$): which query directions are observable through the retained keys. ii) **Value/Output Channels** ($U_C$): how strongly those directions affect the model output. iii) **Query Statistics** ($\Lambda_Q$): which directions are likely to be queried in future decoding steps. iv) **Noise Level** ($\Sigma_{\text{noise}}$): uncertainty arising from modeling approximations. This unified formulation in Eq. (5) provides theoretical insight into the effectiveness of existing KV cache eviction heuristics. For example, methods such as KeyDiff (Park et al., 2025) favor geometrically diverse keys, often measured via cosine dissimilarity, which implicitly enlarges the effective span of $K_C$. Under simplified assumptions—such as isotropic query and noise distributions and ignoring value-channel effects—this behavior corresponds to maximizing a reduced objective of the form $\log \det(I + K_C K_C^\top)$. Similar interpretations apply to other attention-based and diversity-driven eviction strategies (see detailed analysis in Appendix B).

### 3.5. CapKV: Capacity-Inspired KV-Cache Eviction

While the objective in Eq. (5) provides a principled criterion for cache eviction, directly optimizing it during infer-

ence is computationally infeasible. We therefore derive a lightweight approximation that preserves its essential structural properties, leading to our proposed *CapKV* eviction strategy. Specifically, Eq. (5) indicates that information preservation is governed by the effective rank and diversity of the query-to-output channel induced by the retained KV subset. Accordingly, a practical eviction policy inspired by Theorem 3.2 should (i) promote diversity in the induced output directions, (ii) suppress redundancy among retained KV pairs, and (iii) account for their relative importance under likely future queries. To this end, we construct a capacity matrix in the output space,

$$A = I + \sum_{i \in C} w_i \, u_i u_i^\top, \qquad (6)$$

where $u_i$ is the value-induced output direction associated with token $i$, and $w_i \geq 0$ is a query-dependent weight reflecting its relative importance under likely future queries. The identity term ensures numerical stability and can be interpreted as an isotropic noise prior. In practice, we approximate the output direction using the value vector itself, i.e., $u_i = v_i$. This simplification treats the shared output projection as an implicit linear transformation that preserves relative diversity structure, while substantially reducing computational overhead. Although directly maximizing $\log \det(A)$ remains expensive, classical results from D-optimal experimental design provide an efficient approximation of its marginal gains (Lanouette et al., 1997; Mitchell, 1974).

**Lemma 3.3** (Matrix determinant lemma). *Given a positive definite matrix $A$ and a rank-one update $w_i u_i u_i^\top$, the marginal increase in the log-determinant objective is characterized by the following first-order approximation:*

$$\log \det(A + w_i u_i u_i^\top) - \log \det(A) \approx w_i \, u_i^\top A^{-1} u_i.$$

**One-shot eviction score.** Lemma 3.3 characterizes the first-order contribution of an individual KV pair to the capacity objective. The function induced by Eq. (6) is monotonic with diminishing returns, enabling a greedy approximation to the underlying subset selection problem. In the context of cache eviction, this motivates removing KV pairs with the smallest marginal contributions. We therefore assign each KV pair the eviction score

$$s_i = w_i \, u_i^\top A^{-1} u_i. \qquad (7)$$

The remaining design choice concerns the specification of the query-dependent weight $w_i$. In CAPKV, we adopt a lightweight proxy based on historical query statistics,

$$\mu_q = \mathbb{E}[q], \qquad w_i = \exp(k_i^\top \mu_q \cdot \tau). \qquad (8)$$

Here, $\tau \geq 0$ is a tunable hyperparameter that controls the influence of historical query information: $\tau = 0$ corresponds to ignoring query-dependent priors, while larger

values place increasing emphasis on alignment with past queries. Although the capacity objective in Theorem 3.2 depends only on the query covariance and is invariant to the query mean, cache eviction operates on finite, structured query realizations observed online. In this setting, $\mu_q$ serves as a low-variance importance prior that captures dominant query directions encountered so far, complementing the second-order, diversity-driven structure induced by the log-determinant objective.

**Computational complexity.** The full procedure is summarized in Algorithm 1. Let $N$ denote the number of cached key–value pairs and $d$ the value dimension. Computing the mean query and the query-alignment weights scales linearly with $N$ and is fully parallelizable. The dominant computational cost arises from constructing and operating on the capacity matrix in the value space. Over all, the eviction cost scales linearly with the number of cached entries $N$ and quadratically with the value dimension $d$, i.e., $O(Nd^2 + d^3)$. While not independent of the context length, this avoids higher-order dependence on $N$ and remains negligible compared to the overall attention computation cost in long-context decoding. Additional empirical runtime evaluations are provided in Appendix C.1, which show that our method introduces minimal computational overhead, placing it in the same order of magnitude as existing approaches.

---

**Algorithm 1** CapKV: Capacity-Inspired KV-Cache Eviction

1: **Input:** KV cache $\{(k_i, v_i)\}_{i=1}^N$, historical queries $\{q_j\}_{j=1}^T$, cache budget $B$
2: **Output:** Retained index set $C$, $|C| = B$
3: Compute mean query $\mu_q \leftarrow mean(q)$
4: **for** $i = 1$ **to** $N$ **do**
5:     Compute query-alignment weight $w_i$ by Eq. (8)
6:     Set output proxy $u_i \leftarrow v_i$
7: **end for**
8: Compute capacity matrix $A$ by Eq. (6)
9: **for** $i = 1$ **to** $N$ **do**
10:     Compute leverage score $s_i$ by Eq. (7)
11: **end for**
12: Select top-$K$ indices $C \leftarrow \text{TOPK}(s, B)$
13: **Return** $C$

---

## 4. Experiments

We conducted experiments from three complementary perspectives: i) the relationship between the proposed capacity-based objective and downstream performance across methods and datasets, ii) effectiveness under strict KV cache budgets on long-context reasoning benchmarks, and iii) robustness to key hyperparameters. Together, these experiments assess both the practical benefits of CAPKV and the explanatory value of the proposed information-theoretic framework. Our primary experiments are conducted on Qwen3-8B and

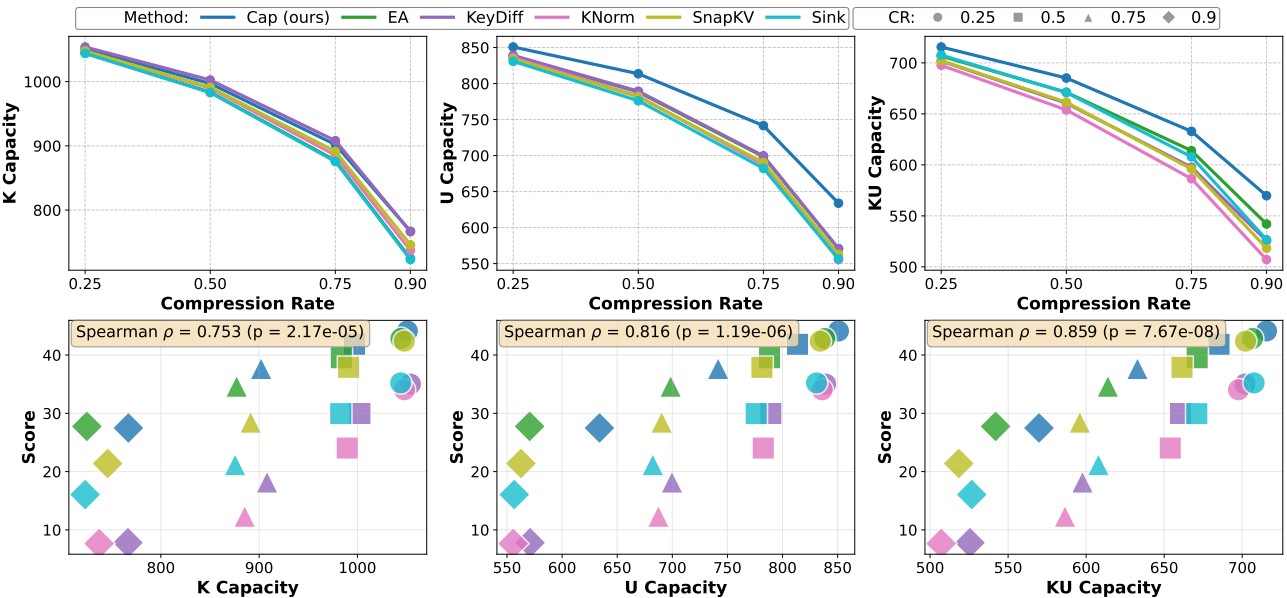

*Figure 3.* Capacity preservation and capacity–performance correlation on **Qasper**.**Top:** Retained capacity under increasing cache compression ratios for different KV cache eviction methods, shown for three simplified capacity proxies.**Bottom:** Scatter plots of retained capacity versus task performance across eviction methods and compression ratios, with Spearman rank correlation coefficients and corresponding $p$-values annotated.Results on additional datasets are provided in Appendix C.2.

Qwen3-14B (Team, 2025) using LongBench (16 tasks) (Bai et al., 2023). We compare CAPKV with representative KV cache eviction baselines, including EA (Devoto et al., 2025), KeyDiff (Park et al., 2025), SnapKV (Li et al., 2024b), Sink (Xiao et al., 2023) and Knorm (Devoto et al., 2024), under identical cache budgets and eviction protocols (see experimental parameter settings in Appendix C).

### 4.1. Does Capacity Predict Performance?

A central claim of our work is that KV cache eviction can be understood as the preservation of effective information capacity under memory constraints. In this section, we empirically examine whether this capacity-based perspective provides a meaningful explanation of downstream performance degradation across different eviction strategies. Specifically, this analysis serves as an empirical validation of the unified framework, by assessing whether methods that better preserve information capacity also tend to maintain higher task performance. We conduct this study on Qwen3-8B across four long-context benchmarks: 2WIKIMQA, MULTIFIELDQA, PASSAGERETRIEVAL, and QASPER. We evaluate six representative KV cache eviction methods under four cache compression ratios (0.25, 0.5, 0.75, and 0.9). For each input instance, we record both the final task score and the retained KV cache at every Transformer layer.

**Capacity proxies.** Directly evaluating the full capacity expression in Theorem 3.2 during inference is impractical, as it depends on the unknown covariance of future queries. To enable a method-agnostic and interpretable analysis, we

therefore introduce three simplified capacity proxies by assuming an isotropic query distribution:

$$K - Capacity = \log\det(I + K_C K_C^\top), \quad (9)$$

$$U - Capacity = \log\det(I + U_C U_C^\top), \quad (10)$$

$$KU - Capacity = \log\det(I + U_C K_C K_C^\top U_C^\top). \quad (11)$$

These proxies isolate different structural components of the unified objective, capturing diversity in the key space, diversity in the output space, and their joint effect, respectively. Importantly, these quantities are used solely as diagnostic indicators of information retention rather than predictive performance metrics. For each retained KV subset, capacity values are computed at every layer and averaged across layers for each input instance. Dataset-level statistics are obtained by further averaging over all inputs.

**Capacity preservation and correlation analysis.** Figure 3 shows the capacity preservation and capacity–performance correlation on Qasper. As compression increases, all methods exhibit a monotonic decrease in retained capacity, while methods achieving stronger downstream performance consistently preserve higher capacity across all three proxies. The bottom row further reveals strong and statistically significant positive correlations between retained capacity and task performance (Spearman $\rho$ ranging from 0.75 to 0.85), indicating a close association between capacity preservation and performance under KV cache eviction. Consistent trends are observed across the remaining datasets (see full results in Appendix C.2). A comparison across capacity proxies further highlights the limitation of eviction strate-

*Table 1.* Experimental results of Qwen3-8B and Qwen3-14B on Longbench Benchmark.

| | | SD-QA | | | MD-QA | | | Summ. | | | FSL | | | Synth. | | Code | | |
|---|---|---|---|---|---|---|---|---|---|---|---|---|---|---|---|---|---|---|
| | C.R. | NQA | Qser | MF | HpQA | 2WQA | Mque | GR | QMS | MN | TREC | TQA | SSum | PC | PR | Lcc | RBP | Avg. |
| Qwen3-8B | | 28.89 | 43.57 | 54.76 | 62.86 | 49.20 | 35.69 | 33.67 | 24.60 | 24.99 | 41.00 | 90.21 | 39.95 | 10.50 | 91.02 | 66.83 | 61.95 | 47.48 |
| EA | 0.25 | 28.69 | 42.84 | 53.02 | 62.23 | 48.37 | 33.99 | **33.61** | 24.22 | 24.94 | 57.00 | 88.13 | 40.25 | 8.00 | 91.81 | 66.33 | 62.17 | 47.85 |
| | 0.5 | 28.34 | 39.54 | **51.11** | **61.22** | **46.78** | 30.13 | 33.38 | 24.00 | **24.94** | 64.00 | 86.96 | 39.81 | 8.50 | 87.46 | 65.41 | 62.30 | 47.12 |
| | 0.75 | 27.23 | 34.65 | 39.60 | 57.52 | 39.27 | 27.58 | **32.04** | 22.76 | **23.70** | **70.00** | 85.29 | 39.71 | **10.00** | 49.74 | 59.30 | 63.46 | 42.62 |
| | 0.9 | 23.28 | 27.76 | 32.42 | 43.62 | 31.74 | 23.28 | **29.65** | 20.98 | **22.00** | **61.75** | 84.16 | 38.15 | **11.50** | 17.42 | 50.93 | 62.92 | 36.35 |
| Keydiff | 0.25 | 25.44 | 35.02 | 49.12 | 47.28 | 42.49 | 23.07 | 32.84 | 22.94 | 24.93 | 62.50 | 83.61 | 38.84 | 6.37 | 91.50 | 61.75 | 56.71 | 44.03 |
| | 0.5 | 23.30 | 29.97 | 39.86 | 35.86 | 33.86 | 16.60 | 29.49 | 22.43 | 21.93 | 55.00 | 85.62 | 38.25 | 5.49 | 86.33 | 46.51 | 53.65 | 39.01 |
| | 0.75 | 14.90 | 18.21 | 29.26 | 19.68 | 27.49 | 9.51 | 21.97 | 21.18 | 14.22 | 39.00 | 81.56 | 36.15 | 5.81 | 45.58 | 27.38 | 53.66 | 29.10 |
| | 0.9 | 10.62 | 7.80 | 23.18 | 15.76 | 23.01 | 7.01 | 16.07 | 19.85 | 8.02 | 6.50 | 78.15 | 31.41 | 7.78 | 13.92 | 13.57 | 52.57 | 20.95 |
| Knorm | 0.25 | 25.35 | 34.04 | 49.59 | 49.24 | 40.66 | 24.62 | 32.53 | 23.17 | 24.51 | 63.00 | 84.55 | **41.94** | **11.50** | 93.00 | 50.21 | 47.71 | 43.48 |
| | 0.5 | 17.84 | 24.04 | 39.55 | 29.63 | 28.61 | 14.82 | 28.85 | 21.55 | 21.26 | 50.50 | 81.30 | 40.80 | 6.64 | 78.46 | 36.13 | 51.59 | 35.72 |
| | 0.75 | 13.39 | 12.33 | 27.76 | 13.90 | 18.25 | 6.39 | 22.63 | 20.24 | 14.73 | 25.75 | 81.36 | 38.51 | 9.88 | 20.00 | 16.83 | 54.66 | 24.79 |
| | 0.9 | 9.66 | 7.65 | 23.48 | 8.78 | 19.54 | 4.11 | 15.46 | 19.24 | 8.15 | 14.00 | 80.68 | 33.08 | 3.75 | 6.50 | 11.54 | 55.83 | 20.09 |
| Snapkv | 0.25 | **28.94** | 42.38 | 51.45 | **63.39** | 48.67 | 34.44 | 33.17 | **24.33** | 24.20 | 39.00 | **89.81** | 41.03 | 10.50 | 91.58 | **67.02** | 61.65 | 46.97 |
| | 0.5 | 26.46 | 37.89 | 45.60 | 58.99 | 45.84 | 33.18 | 31.85 | 23.21 | 23.38 | 40.50 | **88.91** | 40.89 | 8.00 | 90.71 | **68.10** | 62.26 | 45.36 |
| | 0.75 | 25.43 | 28.48 | 35.60 | 52.72 | 38.90 | 26.24 | 29.48 | 21.37 | 21.10 | 38.50 | **89.09** | 40.36 | 6.55 | 88.09 | 67.73 | 63.33 | 42.06 |
| | 0.9 | 21.54 | 21.39 | 26.12 | 42.40 | 29.25 | 19.73 | 25.10 | 19.19 | 17.70 | 30.50 | **87.49** | 38.41 | 9.05 | 46.63 | 65.89 | 64.07 | 35.28 |
| Cap(ours) | 0.25 | 28.30 | **44.13** | **54.26** | 62.95 | 48.47 | **34.93** | 33.55 | 24.21 | **25.05** | **72.00** | 88.47 | 41.00 | 10.50 | **95.54** | 65.60 | **63.23** | **49.51** |
| | 0.5 | **29.43** | 41.84 | **51.11** | 61.15 | 44.40 | **34.08** | **34.25** | **24.25** | 24.59 | **66.50** | 87.49 | 39.64 | **11.00** | **98.42** | 60.06 | **63.32** | **48.15** |
| | 0.75 | **28.62** | **37.69** | **44.16** | 54.33 | 38.44 | **30.54** | 30.99 | **23.58** | 23.12 | 60.50 | 85.67 | 38.29 | 9.50 | **96.96** | 51.92 | **63.81** | **44.88** |
| | 0.9 | **24.60** | 27.49 | 36.12 | **47.69** | 31.49 | **25.12** | 27.11 | **21.61** | 20.27 | 36.00 | 85.97 | 35.00 | 6.50 | **57.38** | 39.65 | 63.06 | **36.57** |
| Qwen3-14B | | 30.32 | 44.05 | 52.14 | 62.30 | 55.87 | 32.79 | 32.84 | 24.32 | 24.96 | 70.00 | 89.10 | 42.01 | 9.80 | 99.42 | 69.36 | 67.27 | 50.41 |
| EA | 0.25 | 29.31 | **44.25** | 51.06 | 63.10 | **56.44** | 32.64 | 32.79 | 24.47 | 24.85 | 70.50 | 89.60 | 41.49 | 9.00 | 97.45 | 69.30 | 66.54 | 50.17 |
| | 0.5 | 28.28 | 43.70 | 49.47 | 63.83 | 50.77 | 31.47 | 32.29 | 24.13 | **25.15** | 70.50 | 90.56 | 41.46 | 8.70 | 94.17 | 67.75 | 65.82 | 49.25 |
| | 0.75 | 23.62 | 35.92 | 37.70 | 52.92 | 38.91 | 26.90 | **31.37** | 22.79 | 24.74 | 66.50 | **90.10** | 41.20 | 8.55 | 72.74 | 61.32 | 64.21 | 43.72 |
| | 0.9 | 22.78 | 25.16 | 27.36 | 41.20 | 23.29 | 20.64 | **28.79** | 21.28 | **22.39** | 61.42 | 89.02 | 39.65 | **10.50** | 27.96 | 50.87 | 62.00 | 35.89 |
| Keydiff | 0.25 | 28.11 | 36.18 | 50.24 | 51.61 | 45.09 | 26.48 | 33.02 | 24.06 | 24.94 | 69.00 | 82.57 | 41.52 | **11.50** | 98.58 | 66.02 | 62.22 | 46.95 |
| | 0.5 | 24.93 | 30.20 | 44.89 | 41.04 | 35.61 | 21.60 | 28.69 | 23.16 | 22.65 | 57.50 | 75.73 | 39.56 | **13.00** | 95.50 | 49.69 | 57.42 | 41.32 |
| | 0.75 | 21.03 | 21.80 | 44.63 | 26.79 | 27.78 | 14.63 | 20.31 | 21.41 | 15.55 | 33.00 | 74.33 | 34.99 | 10.50 | 81.83 | 28.68 | 54.69 | 32.59 |
| | 0.9 | 12.76 | 12.12 | 24.13 | 18.97 | 25.03 | 6.77 | 11.41 | 19.50 | 6.68 | 3.00 | 74.32 | 30.10 | **10.50** | 39.00 | 15.19 | 54.80 | 22.77 |
| Knorm | 0.25 | 28.28 | 40.58 | 52.17 | 54.61 | 43.52 | 28.23 | **33.24** | 23.74 | **24.95** | 65.00 | 79.56 | 39.75 | 10.50 | **100.00** | 58.96 | 62.29 | 46.59 |
| | 0.5 | 23.58 | 34.56 | 44.65 | 43.17 | 36.84 | 15.95 | 26.18 | 22.96 | 22.80 | 60.00 | 78.35 | 39.15 | 12.50 | 95.08 | 44.16 | 59.78 | 41.23 |
| | 0.75 | 17.72 | 17.80 | 35.08 | 22.71 | 25.99 | 6.97 | 15.05 | 20.99 | 16.87 | 44.50 | 82.07 | 36.27 | **13.00** | 63.58 | 27.58 | 54.15 | 31.27 |
| | 0.9 | 10.07 | 16.59 | 26.61 | 11.10 | 21.02 | 4.54 | 10.45 | 19.66 | 11.37 | 23.50 | 77.52 | 33.51 | 8.50 | 13.58 | 20.85 | 46.55 | 22.21 |
| Snapkv | 0.25 | 30.08 | 42.60 | 50.65 | 62.21 | 55.07 | 33.04 | 32.16 | 23.61 | 24.82 | 70.50 | 89.60 | **42.21** | 10.50 | 98.33 | **69.89** | 66.51 | 50.11 |
| | 0.5 | 28.24 | 41.10 | 42.15 | 62.48 | 50.86 | 33.89 | 31.20 | 23.21 | 24.00 | 65.50 | 89.60 | **42.06** | 9.00 | 96.89 | **69.37** | 66.12 | 48.48 |
| | 0.75 | 25.77 | 31.24 | 35.66 | 55.32 | 43.13 | 25.51 | 29.08 | 21.07 | 22.11 | 39.25 | 89.54 | **42.23** | 6.67 | 93.38 | **67.42** | 64.25 | 44.43 |
| | 0.9 | 22.43 | 18.72 | 26.75 | 42.78 | 28.44 | 19.35 | 25.16 | 19.07 | 18.27 | 39.25 | 89.19 | 40.59 | 5.75 | 72.02 | **63.91** | 62.40 | 37.13 |
| Cap(ours) | 0.25 | **30.56** | 43.25 | **52.49** | **63.55** | 54.25 | **35.68** | 32.85 | **24.66** | 24.88 | **75.00** | **90.10** | 41.80 | 8.00 | 99.08 | 69.42 | **67.25** | **50.80** |
| | 0.5 | **29.92** | **43.96** | **51.95** | **65.06** | 54.11 | **35.42** | 32.80 | 24.19 | 24.87 | 72.00 | **91.85** | 41.27 | 9.50 | 97.62 | 66.82 | **67.84** | **50.57** |
| | 0.75 | 27.62 | 37.53 | 45.64 | **58.93** | 46.67 | **32.22** | 31.09 | **24.24** | 23.70 | 66.00 | 89.71 | 40.80 | 7.15 | **98.58** | 57.86 | **66.89** | **47.16** |
| | 0.9 | **25.16** | **27.78** | **35.16** | **49.75** | 32.62 | **26.88** | 27.45 | **21.98** | 20.70 | 48.25 | 87.82 | 38.41 | 5.50 | **76.67** | 45.08 | **63.25** | **39.53** |

gies that emphasize diversity in a single representational space. Approaches like KeyDiff tend to preserve relatively higher *K-Capacity* by promoting geometric diversity among keys, yet this advantage does not consistently translate into comparable preservation of *U-Capacity* or *KU-Capacity*, nor into proportional gains in downstream performance. This gap reflects the fact that effective information capacity arises from the joint interaction between keys and values, rather than from diversity in either space alone. In contrast, eviction strategies that account for this coupling exhibit more consistent capacity preservation across all proxies, aligning more closely with downstream performance. All these observations support the view that KV cache eviction cannot be reduced to optimizing a single notion of diversity or importance, but instead requires a unified capacity-based perspective that balances multiple interdependent factors.

## 4.2. Results on Long-context Benchmark

Having established that preserving effective information capacity is strongly associated with downstream performance, we now evaluate the practical benefits of explicitly optimizing this objective through CAPKV. We conduct comprehensive experiments on LONGBENCH, a widely used benchmark comprising 16 long-context tasks spanning question answering, summarization, reasoning, and code understanding. Our primary evaluations are performed on Qwen3-8B and Qwen3-14B. Additional evaluations on other model architectures, including Llama3.1-8B (Team, 2024), Mistral-7B (Jiang et al., 2023), and Qwen3-4B, are provided in Appendix C.3. Table 1 summarizes the performance on LongBench. Results are grouped by task category for clarity, with each entry representing the average score over the corresponding datasets. As shown in Table 1, CAPKV

consistently achieves superior performance under KV cache compression on both Qwen3-8B and Qwen3-14B. Across all evaluated compression ratios, CAPKV outperforms all baseline methods in terms of average score. Similar performance trends are also observed on additional model architectures.

On question answering (QA) tasks, CAPKV better preserves accuracy under compression, reflecting its ability to retain diverse and complementary contextual signals required to support varied future queries. For summarization and code tasks, CAPKV exhibits more graceful performance degradation as cache budgets shrink, avoiding the sharp drops observed with heuristic eviction strategies. These results indicate that explicitly balancing multiple sources of information—rather than prioritizing a single notion of importance or diversity—yields more robust behavior across heterogeneous long-context workloads.

### 4.3. Results on Reasoning Benchmark

Beyond prefill-phase cache eviction, practical long-context inference often requires managing the KV cache dynamically during autoregressive decoding. This setting poses a more stringent test for eviction strategies, as errors introduced early in decoding can accumulate and significantly degrade final performance. We evaluate CAPKV in this decoding-phase eviction setting using the NEMOTRON-7B reasoning model (Ahmad et al., 2025) on the AIME25 benchmark (Zhang & Math-AI, 2025). All methods perform cache eviction every 512 decoding steps. When the number of generated tokens exceeds the reserved cache budget, KV entries with lower eviction scores are removed. We note that SNAPKV is not included in this comparison, as it relies on computing window-based attention statistics that require real-time window construction, which is not feasible in the decoding-phase eviction setting.

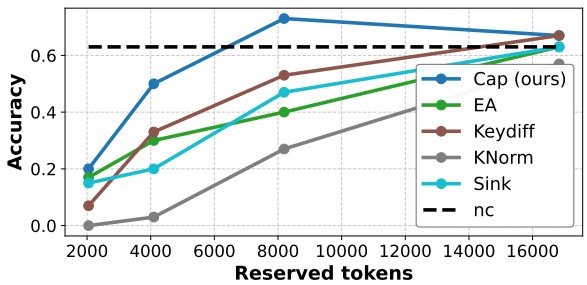

*Figure 4.* Nemotron7B Reasoning Model Experiments on Aime25. nc stands for no compression.

Figure 4 reports reasoning accuracy as a function of the reserved cache size. Notably, CAPKV exhibits notably more stable and graceful degradation as cache budgets shrink. While heuristic eviction strategies based on local importance signals tend to suffer sharper performance drops in this irreversible setting, CAPKV maintains a smoother accu-

racy–budget curve, indicating greater robustness to eviction errors introduced early in decoding. This behavior aligns with the capacity-based perspective: by preserving a balanced and diverse set of KV entries, CAPKV better supports the evolving distribution of future queries throughout long reasoning chains. These results demonstrate that capacity-aware eviction remains effective even in the most challenging inference regimes, where mistakes cannot be corrected and robustness is essential.

### 4.4. Ablation Study

We evaluate two pivotal approximation designs in CAPKV: the output-direction proxy ($u_i = v_i$) and the query-aware gating mechanism ($k_i^\top \mu_q$). Specifically, we evaluate three variants using Qwen3-8B on five representative LongBench subsets: (i) default CAPKV, (ii) CAPKV with exact output direction ($u_i = W_O v_i$), and (iii) CAPKV with a diagonal covariance-based gating statistic ($k_i^\top \text{Diag}(\Sigma_q) k_i$).

*Table 2.* Ablation study on the output-direction proxy and query-aware gating design using Qwen3-8B on LongBench subsets.

| Method | C.R. | 2WQA | MN | MF | PR | TREC | Avg. |
|---|---|---|---|---|---|---|---|
| CAP | 0.5 | 44.40 | **24.59** | 51.11 | **98.42** | 66.50 | 57.00 |
| W/ Exact $u$ | 0.5 | 43.76 | 24.76 | **51.88** | 98.02 | **67.50** | **57.18** |
| W/ Cov. | 0.5 | **47.26** | 23.98 | 50.29 | 97.06 | 64.00 | 56.52 |
| CAP | 0.75 | 38.44 | 23.12 | 44.16 | 96.96 | **60.50** | 52.64 |
| W/ Exact $u$ | 0.75 | 39.29 | **23.35** | **44.18** | **97.54** | 60.00 | **52.87** |
| W/ Cov. | 0.75 | **39.71** | 21.84 | 43.63 | 94.92 | 46.00 | 49.22 |

As shown in Table 2, invoking the exact output direction $u_i = W_O v_i$ brings negligible marginal improvements ($< 0.23\%$ on average) across both compression ratios. However, explicitly computing $W_O v_i$ scales the complexity of capacity matrix construction from $\mathcal{O}(Nd^2 + d^3)$ to $\mathcal{O}(N(d \cdot h)^2 + (d \cdot h)^3)$, where $h$ is the number of attention heads. Consequently, the identity proxy $u_i = v_i$ strikes a much more favorable efficiency–performance trade-off and serves as our default.

For the query-aware gating term, replacing the mean-based prior with the covariance-based counterpart incurs a slight performance degradation. This does not contradict our theoretical formulation where query covariance guides the optimal capacity; rather, it highlights the practical difficulty of accurately estimating second-order statistics from limited, finite historical queries during inference. The empirical covariance introduces substantial noise under aggressive compression, whereas the mean-based statistic acts as a robust, low-variance prior that stabilizes eviction decisions.

We further investigate the sensitivity of CAPKV to the temperature parameter $\tau$, which controls the strength of query-aware weighting. This study assesses whether incorporating query relevance meaningfully complements the diversity-driven capacity objective, rather than serving as a fragile

*Table 3.* Performance comparison under different $\tau$ values.

| C.R. | $\tau = 0$ | $\tau = 1$ | $\tau = 5$ | $\tau = 7$ | $\tau = 10$ |
|------|--------|--------|--------|--------|---------|
| 0.25 | 47.61 | 47.66 | **47.94** | 47.02 | 45.85 |
| 0.5 | 46.07 | 46.71 | **46.91** | 46.79 | 45.92 |
| 0.75 | 38.49 | 39.34 | **39.95** | 39.46 | 37.93 |
| 0.9 | 30.19 | 32.05 | 32.11 | 30.41 | **32.41** |
| Average | 40.59 | 41.44 | **41.73** | 40.92 | 40.53 |

tuning component. Experiments are conducted on Llama 3.1–8B using LongBench, with cache compression ratios ranging from 0.25 to 0.9. Table 3 summarizes the results under different values of $\tau$. Setting $\tau = 0$ corresponds to a query-agnostic variant. Overall, moderate values of $\tau$ perform best: $\tau = 5$ achieves the highest average score and is consistently effective across compression settings. In contrast, large $\tau$ values degrade performance, particularly under aggressive compression, due to over-emphasizing a small subset of keys and reducing cache diversity.

### 4.5. Compatibility with Quantization

*Table 4.* Compatibility with 4-bit HQQ KV-cache quantization.

| C.R. = 0.5 | 2WQA | Lcc | MN | MF | PR | TREC | Avg. |
|------------|------|------|------|------|------|------|------|
| EA | **46.78** | 65.41 | **24.94** | **51.11** | 87.46 | 64.00 | 55.14 |
| KeyDiff | 33.86 | 46.51 | 21.93 | 39.86 | 86.33 | 55.00 | 45.70 |
| KNorm | 28.61 | 36.13 | 21.26 | 39.55 | 78.46 | 50.50 | 40.80 |
| SnapKV | 45.84 | **68.10** | 23.38 | 45.60 | 90.71 | 40.50 | 54.73 |
| CAP(Ours) | 44.54 | 59.47 | 24.47 | 51.02 | **97.85** | **66.50** | **55.47** |

| C.R. = 0.75 | 2WQA | Lcc | MN | MF | PR | TREC | Avg. |
|-------------|------|------|------|------|------|------|------|
| EA | 39.27 | 59.30 | **23.70** | 39.60 | 49.74 | **70.00** | 42.32 |
| KeyDiff | 27.49 | 27.38 | 14.22 | 29.26 | 45.58 | 39.00 | 28.79 |
| KNorm | 18.25 | 16.83 | 14.73 | 27.76 | 20.00 | 25.75 | 19.51 |
| SnapKV | 38.90 | **67.73** | 21.10 | 35.60 | 88.09 | 38.50 | 50.28 |
| CAP(Ours) | **39.27** | 51.51 | 23.21 | **44.72** | **97.58** | 60.50 | **51.26** |

KV cache eviction and KV cache quantization are complementary techniques for reducing the memory footprint of long-context inference. While eviction reduces the number of retained KV entries, quantization further compresses the numerical representation of the remaining cache. We therefore evaluate whether CAPKV remains effective when combined with low-bit KV-cache quantization. Specifically, we apply 4-bit HQQ quantization (Badri & Shaji, 2023) to the retained KV cache during inference and evaluate different eviction methods on Qwen3-8B using LongBench subsets under compression ratios of 0.5 and 0.75.

As shown in Table 4, CAPKV remains effective when combined with 4-bit HQQ KV-cache quantization and achieves the best average performance at both compression ratios. These results suggest that the capacity-aware eviction criterion is compatible with low-bit KV-cache quantization. In

particular, CAPKV preserves its advantage even when the retained cache is subject to quantization noise, indicating that the selected KV entries remain informative and robust under additional numerical compression.

### 4.6. Results under Ultra-long Contexts

*Table 5.* Results on LongBench v2 under ultra-long contexts. The uncompressed baseline score is 44.5.

| C.R. | EA | KeyDiff | KNorm | SnapKV | Cap(Ours) |
|------|------|---------|--------|--------|-----------|
| 0.5 | 45.4 | 29.4 | 31.9 | **46.2** | 43.7 |
| 0.75 | 45.4 | 31.1 | 23.5 | 38.7 | **47.9** |
| 0.9 | 43.7 | 22.7 | 16.8 | 42.9 | **44.8** |

To verify the scalability of CAPKV in extreme scenarios, we evaluate its performance on LongBench–v2 (Bai et al., 2024) using 118 samples with context lengths spanning 32K to 128K. Since these sequences exceed the native window of Qwen3-8B, we employ YaRN (Peng et al., 2026) extrapolation to extend the context length. We compare various KV cache eviction methods under cache compression ratios of 0.5 to 0.9. As shown in Table 5, CAPKV remains robust under ultra-long contexts. In particular, under high compression ratios of 0.75–0.9, CAPKV consistently outperforms other eviction methods and achieves performance close to the uncompressed baseline, suggesting that capacity-aware selection remains effective when handling substantially longer contexts.

## 5. Conclusion

We rethink KV cache eviction from an information-theoretic view, framing the process as the preservation of effective information capacity under strict memory constraints. Under a linear–Gaussian surrogate, this perspective yields a unified log-determinant capacity objective that rationalizes existing heuristics as distinct approximations of a common information-preservation goal. Based on this framework, we proposed CAPKV, an efficient capacity-aware method that consistently outperforms prior approaches on long-context benchmarks, providing a more principled and effective path for long-context inference.

**Limitations and Future Work.** The proposed capacity-based framework relies on a linear-Gaussian surrogate, which may not fully capture the nonlinear dynamics in Transformers. Exploring richer surrogate models that incorporate controlled nonlinearity remains an important direction for future work. More broadly, our formulation is intended as a unifying lens rather than a closed-form solution, opening up a broader design space for cache eviction beyond the specific approximation adopted in CAPKV.

## Acknowledgments

This work is supported by the National Major Scientific Instruments and Equipments Development Project of National Natural Science Foundation of China under Grant 62427820, the Sichuan Science and Technology Program under Grant 2025ZDZX0125, the Science Fund for Creative Research Groups of Sichuan Province Natural Science Foundation under Grant 2024NSFTD0035, and the Ministry of Education Engineering Research Center Guiding Project for Machine Learning and Industrial Intelligence Applications under Grant SCU2024D013.

## Impact Statement

This paper presents work whose goal is to advance the field of machine learning. There are many potential societal consequences of our work, none of which we feel must be specifically highlighted here.

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

## A. Detailed Proof of Theorem 3.2

In this appendix, we provide a detailed derivation of the closed-form expression in Theorem 3.2. Throughout the proof, the retained KV subset $Z_C$ is treated as fixed; thus $I(q; Y \mid Z_C)$ should be interpreted as mutual information conditioned on a given realization $Z_C = z_C$.

**Gaussian entropy formula.** If $X \in \mathbb{R}^m$ and $X \sim \mathcal{N}(\mu, \Sigma)$ with $\Sigma \succ 0$, then

$$H(X) = \frac{1}{2} \log \left( (2\pi e)^m \det \Sigma \right). \tag{12}$$

Importantly, the entropy depends only on the covariance $\Sigma$ and is invariant to the mean $\mu$.

**Problem Setup.** Under Assumption 3.1, conditioned on the retained KV subset $Z_C$, the attention-induced output is approximated by the linear-Gaussian model

$$Y = U_C K_C q + \varepsilon,$$

where $q \sim \mathcal{N}(\mu_Q, \Lambda_Q)$, $\varepsilon \sim \mathcal{N}(0, \Sigma_{\text{noise}})$, $q \perp\!\!\!\perp \varepsilon$, and $\Sigma_{\text{noise}} \succ 0$. For convenience, define the effective channel matrix

$$A := U_C K_C,$$

so that $Y = Aq + \varepsilon$.

**Proof.** Conditioned on $q$, $Y$ is Gaussian with

$$Y \mid (q, Z_C) \sim \mathcal{N}(Aq, \Sigma_{\text{noise}}).$$

Marginalizing over $q$, we obtain

$$Y \mid Z_C \sim \mathcal{N}\big(A\mu_Q,\ A\Lambda_Q A^\top + \Sigma_{\text{noise}}\big),$$

where we used independence of $q$ and $\varepsilon$ and linearity of covariance. The query mean $\mu_Q$ affects only the mean of $Y$ and will not influence mutual information.

By definition (with fixed $Z_C$),

$$I(q; Y \mid Z_C) = H(Y \mid Z_C) - H(Y \mid q, Z_C).$$

Since $Y \mid (q, Z_C) \in \mathbb{R}^m$ is Gaussian with covariance $\Sigma_{\text{noise}}$, applying (12) yields

$$H(Y \mid q, Z_C) = \frac{1}{2} \log \left( (2\pi e)^m \det \Sigma_{\text{noise}} \right).$$

Similarly, $Y \mid Z_C$ is Gaussian with covariance $A\Lambda_Q A^\top + \Sigma_{\text{noise}}$, and

$$H(Y \mid Z_C) = \frac{1}{2} \log \left( (2\pi e)^m \det(A\Lambda_Q A^\top + \Sigma_{\text{noise}}) \right).$$

Subtracting the two entropies cancels the $(2\pi e)^m$ factor:

$$\begin{aligned}
I(q; Y \mid Z_C) &= \frac{1}{2} \log \frac{\det(A\Lambda_Q A^\top + \Sigma_{\text{noise}})}{\det \Sigma_{\text{noise}}} \\
&= \frac{1}{2} \log \det \left( \Sigma_{\text{noise}}^{-1}(A\Lambda_Q A^\top + \Sigma_{\text{noise}}) \right) \\
&= \frac{1}{2} \log \det \left( I + \Sigma_{\text{noise}}^{-1} A\Lambda_Q A^\top \right),
\end{aligned}$$

Substituting $A = U_C K_C$ completes the proof:

$$I(q; Y \mid Z_C) = \frac{1}{2} \log \det \left( I + \Sigma_{\text{noise}}^{-1} U_C K_C \Lambda_Q K_C^\top U_C^\top \right). \tag{13}$$

**Discussion.** Equation (13) characterizes the effective information capacity of the linear-Gaussian surrogate channel induced by the retained KV subset. Under a fixed memory budget, maximizing this quantity corresponds to maximizing the predictive information preserved from future queries, which motivates the capacity-based KV cache eviction criterion adopted in this work.

## B. Analysis for Existing Mainstream Methods

This appendix provides an interpretive analysis of representative KV cache eviction methods through the lens of the unified information-theoretic objective introduced in Section 3. In contrast to Appendix A, which presents a formal derivation under a linear-Gaussian surrogate, the goal here is to understand how various heuristic strategies relate to different structural components or approximations of the same information-preservation principle.

All connections drawn in this appendix rely on simplifying assumptions and should be interpreted qualitatively. We do not claim that existing methods explicitly optimize the capacity objective in general Transformer architectures.

### B.1. Recap of the Unified Capacity Objective

Under Assumption 3.1, the information preserved by a retained KV subset $Z_C$ is quantified by

$$I(q; Y \mid Z_C) = \frac{1}{2} \log \det \left( I + \Sigma_{\text{noise}}^{-1} U_C K_C \Lambda_Q K_C^\top U_C^\top \right).$$

This expression highlights four interacting factors: (i) the span and diversity of the retained keys $K_C$, (ii) the value-induced output directions $U_C$, (iii) the statistics of future queries encoded by $\Lambda_Q$, and (iv) modeling uncertainty captured by $\Sigma_{\text{noise}}$. Different eviction strategies can be viewed as prioritizing or approximating different subsets of these factors.

### B.2. Key-Space Diversity Methods

A broad class of KV cache eviction strategies operates purely in the key space, without explicitly considering value magnitudes or output activations. Representative example is Knorm (Devoto et al., 2024), which prioritizes keys with large $\ell_2$ norms, and KeyDiff (Park et al., 2025), which promotes geometric dissimilarity among keys. Although motivated differently, both methods can be interpreted as approximating different structural aspects of the same capacity objective.

**First-order approximation: key-norm heuristics.** Consider a simplified setting in which $\Lambda_Q = \sigma^2 I$, $U_C = I$, and $\Sigma_{\text{noise}} = I$. In this regime, the capacity reduces to

$$\log \det \left( I + K_C K_C^\top \right).$$

A crude first-order approximation ignores interactions between different keys and yields

$$\log \det \left( I + K_C K_C^\top \right) \approx \sum_{i \in C} \|k_i\|^2.$$

From this perspective, Knorm can be viewed as retaining key directions with large marginal channel gain. However, this approximation neglects redundancy: multiple high-norm keys aligned along similar directions may contribute little additional capacity.

This approximation is valid only under restrictive conditions, such as low inter-key correlation or near-orthogonality among retained keys. In regimes where multiple high-norm keys are strongly aligned, the approximation becomes inaccurate, and Knorm may significantly overestimate the true capacity contribution. We therefore emphasize that this connection should be interpreted as a qualitative first-order intuition, rather than a precise optimization equivalence.

**Second-order effects: diversity-based heuristics.** KeyDiff addresses this limitation by explicitly discouraging similarity between retained keys. Geometrically, this corresponds to suppressing large inner products $k_i^\top k_j$, which enter the capacity objective through higher-order interaction terms. In the same simplified regime, the determinant $\det(K_C K_C^\top)$ equals the squared volume spanned by the selected keys, and maximizing it penalizes collinearity and promotes coverage of the key space. KeyDiff can therefore be interpreted as a heuristic that partially accounts for second-order redundancy effects ignored by norm-only criteria.

## B.3. Attention-Based and Query-Aware Methods

A second class of KV cache eviction strategies leverages observed attention patterns or historical queries to guide retention decisions, including SnapKV (Li et al., 2024b) and H2O (Zhang et al., 2023). These approaches prioritize tokens that have received high attention weights during past decoding steps.

**Connection to query statistics.** In the unified framework, the influence of future queries is captured by the query covariance $\Lambda_Q$. Attention weights depend on the inner product $q^\top k_i$, and high attention values indicate strong alignment between observed queries and key directions. Averaging attention scores over time therefore provides an empirical estimate of which key directions are frequently queried, serving as a data-driven proxy for the dominant eigenspaces of $\Lambda_Q$.

**Extreme-case analysis.** Consider a simplified setting where attention weights are approximated by $\exp(q^\top k_i)$ and queries are drawn i.i.d. from a distribution with covariance $\Lambda_Q$. In this regime, the expected attention assigned to key $k_i$ satisfies

$$\mathbb{E}_q\big[\exp(q^\top k_i)\big] \propto \exp\!\big(k_i^\top \mu_Q + \tfrac{1}{2}k_i^\top \Lambda_Q k_i\big),$$

up to normalization. For analytical simplicity, the discussion below focuses on the zero-mean case $\mu_Q = 0$, under which the expected attention weight depends solely on the query covariance $\Lambda_Q$. Thus, keys aligned with high-variance query directions receive systematically larger expected attention, explaining why attention-based eviction methods tend to preserve keys that are most relevant under the empirical query distribution.

We note that the assumption that the noise term $\varepsilon$ is independent of the query is a simplification introduced for analytical tractability. In practice, linearization errors of the softmax and interactions across attention heads may introduce query-dependent residuals. These effects are absorbed into $\Sigma_{\text{noise}}$ in our surrogate model and are not explicitly modeled. Consequently, the resulting capacity objective should be interpreted as a first-order abstraction that captures dominant structural factors, rather than a precise characterization of attention dynamics.

**Complementarity with diversity-based criteria.** Unlike key-diversity or key-norm heuristics, attention-based methods do not explicitly penalize redundancy among retained keys. Multiple keys aligned with the same dominant query direction may all receive high attention scores. This highlights the complementary nature of query-aware and diversity-driven criteria, and motivates hybrid approaches that balance relevance under likely queries with coverage of the key space.

## B.4. A Small-Noise Regime and Effective Rank

We conclude with a limiting analysis that further clarifies the role of diversity.

**Proposition B.1** (Small-noise regime). *Assume $\Lambda_Q = \sigma^2 I$ and $\Sigma_{noise} = \epsilon I$. Let $\{\lambda_j\}$ denote the eigenvalues of $AA^\top$ with $A = U_C K_C$. Then*

$$I(q; Y \mid Z_C) = \frac{1}{2} \sum_j \log\!\Big(1 + \tfrac{\sigma^2}{\epsilon}\lambda_j\Big).$$

*In the limit $\epsilon \to 0$, only directions with $\lambda_j > 0$ contribute, and the capacity is dominated by the number and diversity of non-negligible eigen-directions of the induced channel.*

This analysis provides a formal explanation for why redundancy reduction becomes critical under tight capacity or low-noise regimes.

# C. More experimental results

All experiments were conducted using $4\times$ NVIDIA L40S GPUs. During experiments, we set the CAPKV parameter $\tau$ to 5. The parameter $\tau$ was fixed globally and not tuned per dataset or compression ratio. All baseline methods were evaluated using their recommended default settings from the original implementations. The compression ratio was uniformly fixed across methods to ensure a fair performance comparison.

## C.1. Runtime Efficiency Evaluation

We evaluate the runtime efficiency of CAPKV and representative KV cache eviction baselines on the Qwen3-8B model under varying context lengths and compression ratios. Specifically, we test six methods with input context lengths of 8k,

16k, 32k, and 64k tokens, and compression ratios of 0.2, 0.4, 0.6, and 0.8. For each configuration, the model generates 100 tokens. We perform one warm-up run to eliminate initialization overhead, followed by 5 timed runs, and report the average total generation time. The measured runtime includes both attention computation and cache eviction overhead.

The evaluation results are shown in Fig 5. Across all compression ratios, we observe consistent trends in runtime behavior. The total generation time of all methods increases approximately linearly with the input context length, indicating that KV cache eviction does not alter the overall scaling characteristics of autoregressive decoding. Importantly, CapKV exhibits runtime performance comparable to existing eviction methods across all evaluated settings, including long-context regimes and aggressive compression. Despite involving leverage-score–based selection and lightweight matrix operations, CapKV does not introduce noticeable runtime overhead in practice and remains in the same efficiency range as attention-based and heuristic baselines.

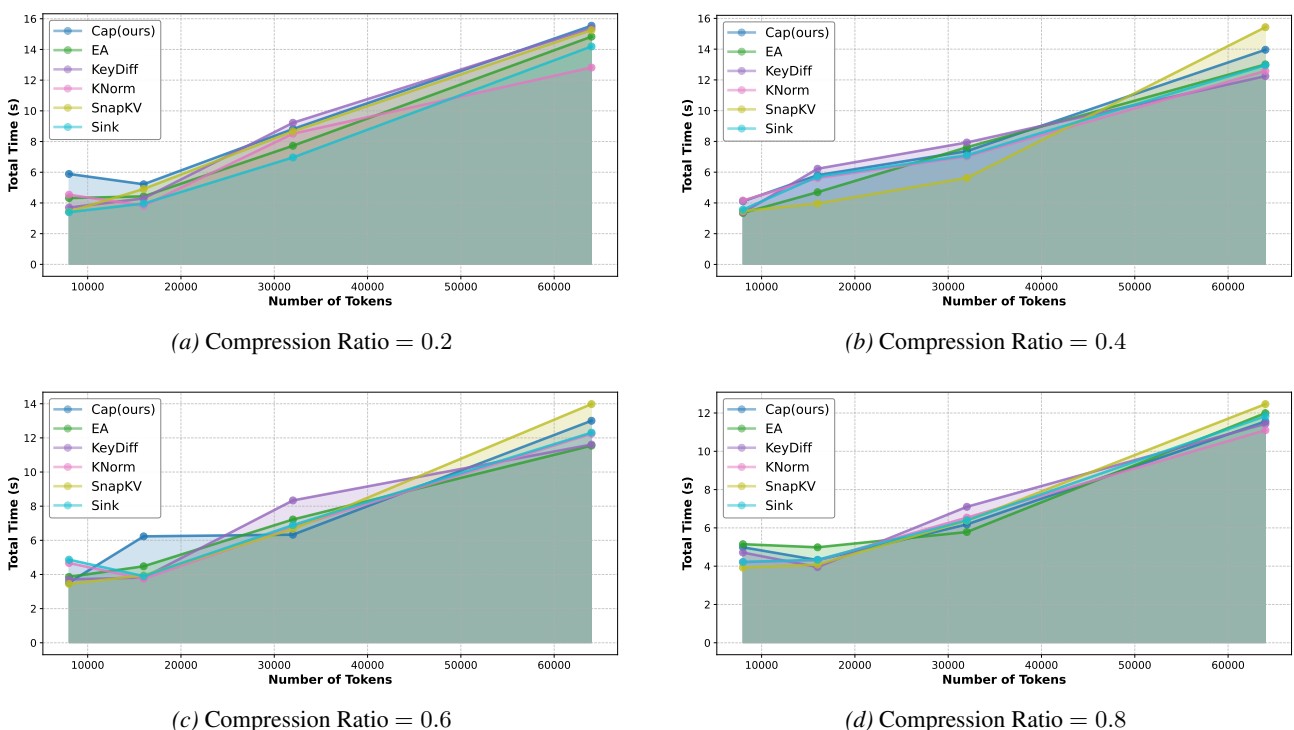

*(a)* Compression Ratio $= 0.2$              *(b)* Compression Ratio $= 0.4$

*(c)* Compression Ratio $= 0.6$              *(d)* Compression Ratio $= 0.8$

*Figure 5.* Runtime comparison of KV cache eviction methods under varying context lengths and compression ratios on Qwen3-8B.

## C.2. Capacity–Performance Correlation

In the main paper, we analyze the relationship between retained KV cache capacity and downstream performance on QASPER. In this section, we extend the same analysis to three additional long-context benchmarks: 2WIKIMQA, MULTIFIELDQA, and PASSAGE-RETRIEVAL. The results are summarized in Figure 6.

**Experimental setup.** The experimental protocol follows Section 4.1 exactly. We evaluate six representative KV cache eviction methods under four compression ratios (0.25, 0.5, 0.75, and 0.9). For each retained KV subset, we compute three simplified capacity proxies and average them across layers and inputs. Task performance is measured using the standard dataset-specific metrics. Spearman rank correlation is reported to quantify the association between retained capacity and downstream performance.

Across all three datasets, the top rows of Figure 6 show a consistent and monotonic decrease in retained capacity as compression becomes more aggressive. Despite differences in task characteristics and input structure, methods that achieve stronger downstream performance consistently preserve higher capacity across all three proxies. In particular, our capacity-aware method(CapKV) maintain a clear advantage under high compression, while more local or heuristic approaches exhibit a sharper degradation in retained capacity.

The bottom rows of Figure 6 report scatter plots of retained capacity versus task performance across eviction methods and compression ratios. We observe strong and statistically significant positive correlations on all three datasets. Specifically, the Spearman correlation coefficients range from approximately $0.73$ to $0.87$, with consistently low $p$-values across all capacity proxies.

These results closely mirror the trends observed on QASPER in the main paper. Despite substantial differences among datasets, the relationship between retained capacity and downstream performance remains robust. This consistency supports the generality of the proposed capacity-based perspective: while the capacity proxies are simplified and diagnostic in nature, they capture structural properties of the KV cache that are strongly aligned with practical performance under memory constraints. The experiment results further strengthen the empirical evidence that effective information capacity provides a unifying explanatory lens for KV cache eviction behavior across tasks and datasets.

### C.3. LongBench Experiments

Beyond the main evaluations on Qwen3-8B and Qwen3-14B, we conduct experiments on several additional widely used open-weight models, including Llama 3.1–8B, Mistral–7B, and Qwen3–4B. All models are evaluated on the LongBench benchmark under identical cache eviction protocols and compression ratios. The corresponding results are summarized in Tables 6, 7, and 8.

Across all evaluated architectures, CAPKV consistently achieves the best or near-best average performance under KV cache compression. In particular, under moderate compression ratios (0.25 and 0.5), CAPKV closely matches or even exceeds the performance of heuristic baselines, while under more aggressive compression, it exhibits sightly smaller performance degradation. Taken together, the additional results reported in this section confirm that the advantages of CAPKV are not specific to a particular model size or architecture. Instead, the method generalizes well across diverse Transformer backbones and inference settings. These findings further support the claim that capacity-aware KV cache eviction provides a principled and effective approach for long-context inference under strict memory constraints.

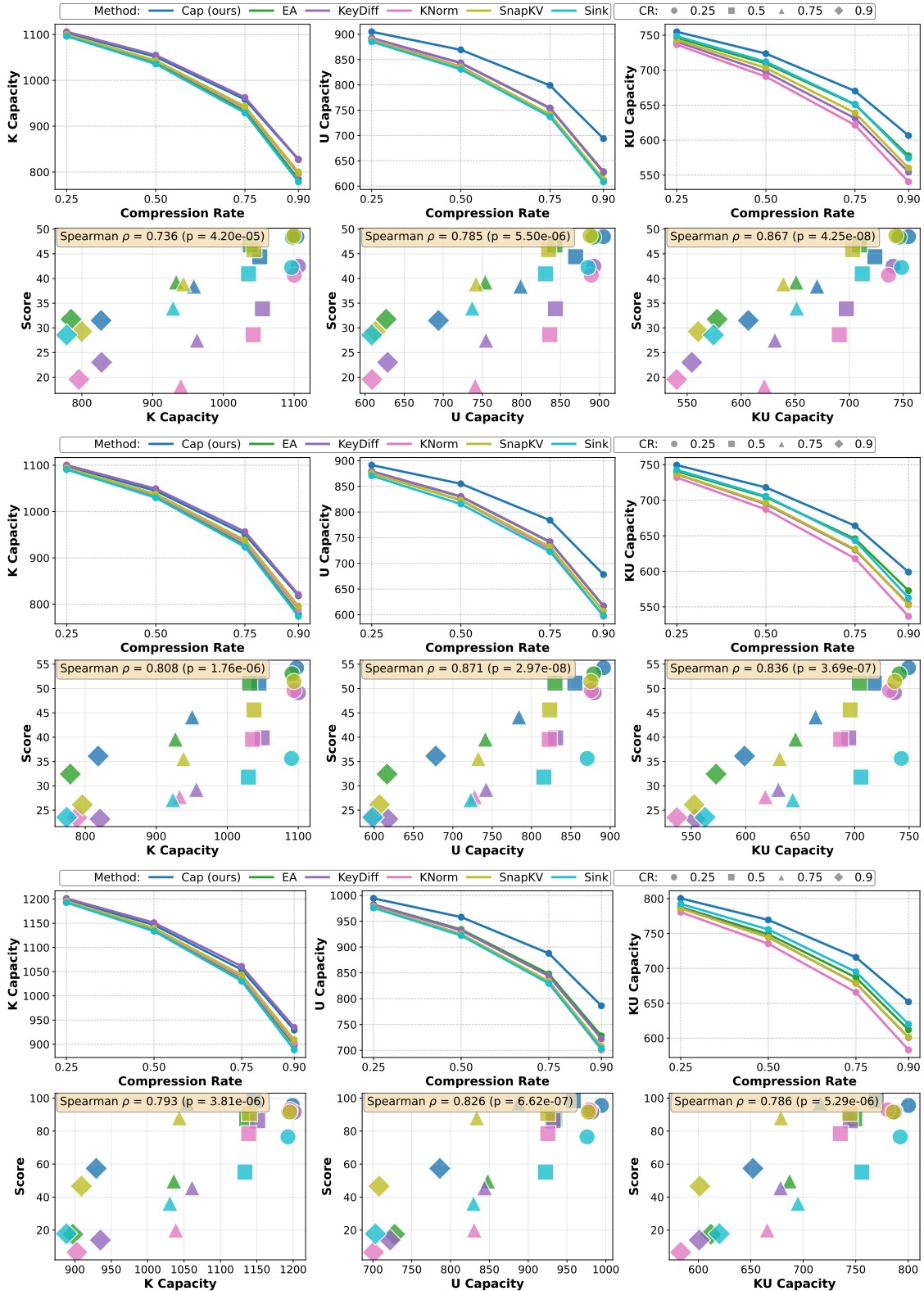

*Figure 6.* Capacity–Performance Correlation analysis on 2WIKIMQA (top two rows), MULTIFIELDQA (middle two rows), and PASSAGE-RETRIEVAL (bottom two rows). For each dataset, the top row shows the retained KV cache capacity under increasing compression ratios for different eviction methods, evaluated using three simplified capacity proxies (*K*, *U*, and *KU-Capacity*). The bottom row plots downstream task performance against the corresponding retained capacity across methods and compression ratios, with Spearman rank correlation coefficients and associated *p*-values annotated.

*Table 6.* Experimental results of Llama3.1-8B on Longbench Benchmark.

| | C.R. | SD-QA | | | MD-QA | | | SUMM. | | | FSL | | | SYNTH. | | CODE | | AVG. |
|---|---|---|---|---|---|---|---|---|---|---|---|---|---|---|---|---|---|---|
| | | NQA | QSER | MF | HpQA | 2WQA | MQUE | GR | QMS | MN | TREC | TQA | SSUM | PC | PR | LCC | RBP | |
| LLAMA3.1-8B | | 30.46 | 47.27 | 55.98 | 59.00 | 51.23 | 33.55 | 35.24 | 25.27 | 26.80 | 29.50 | 86.01 | 39.31 | 10.65 | 100.00 | 53.47 | 47.73 | 45.72 |
| EA | 0.25 | 31.17 | 47.49 | 56.85 | 58.33 | 49.98 | 33.82 | **34.94** | 24.78 | **27.18** | 29.50 | 85.66 | 38.03 | 11.20 | 99.00 | 52.49 | 47.54 | 45.50 |
| | 0.5 | 31.19 | **46.67** | 51.49 | 56.34 | 48.65 | **33.48** | 33.63 | **24.74** | **27.04** | 19.00 | 85.91 | 38.87 | **12.50** | 87.00 | 52.97 | 48.64 | 43.63 |
| | 0.75 | **30.17** | 39.70 | 41.55 | **55.53** | 44.55 | 26.75 | **31.77** | 23.26 | **26.06** | 27.00 | 88.98 | 38.59 | **10.00** | 45.00 | 49.84 | 50.01 | 39.30 |
| | 0.9 | 25.87 | **29.04** | 32.51 | 48.46 | 31.27 | 24.48 | 29.07 | 22.72 | 24.60 | 50.50 | **92.17** | 26.90 | 7.55 | 18.00 | 40.11 | 48.94 | 34.51 |
| KEYDIFF | 0.25 | 31.50 | 47.20 | 54.17 | **58.79** | 50.95 | **35.85** | 34.48 | 24.77 | 26.83 | 62.50 | 86.56 | **40.13** | 10.65 | 99.50 | 52.25 | 47.03 | 47.70 |
| | 0.5 | **32.30** | 44.93 | **52.98** | 55.52 | 46.60 | 30.26 | 32.45 | 24.47 | 25.71 | 64.00 | 87.29 | 40.35 | 10.65 | **99.50** | 45.18 | 46.77 | 46.18 |
| | 0.75 | 29.62 | 33.01 | 41.37 | 52.51 | 37.97 | 25.28 | 29.41 | **23.95** | 22.93 | **52.00** | 84.36 | 40.03 | 9.66 | **99.50** | 37.48 | 47.58 | **41.67** |
| | 0.9 | **28.49** | 16.95 | **34.22** | 40.71 | 25.38 | 15.27 | 26.30 | 21.03 | 19.15 | 40.00 | 80.01 | **39.18** | 8.50 | **89.00** | 26.07 | 47.99 | **34.89** |
| KNORM | 0.25 | 30.71 | 45.34 | 56.16 | 58.43 | 50.63 | 35.27 | 34.07 | **24.83** | 26.67 | 66.00 | 87.26 | 36.85 | 11.15 | 97.50 | 36.91 | 48.35 | 46.63 |
| | 0.5 | 27.45 | 39.59 | 49.46 | 55.72 | 44.23 | 26.95 | 32.22 | 24.25 | 25.51 | 59.00 | 86.44 | 33.60 | 9.55 | 87.50 | 33.05 | 49.16 | 42.73 |
| | 0.75 | 22.18 | 29.67 | 39.44 | 47.45 | 30.45 | 20.51 | 28.68 | 22.78 | 23.48 | 51.50 | 72.85 | 28.98 | 9.15 | 52.50 | 27.20 | 49.17 | 34.75 |
| | 0.9 | 22.44 | 16.23 | 28.48 | 37.69 | 21.73 | 13.30 | 26.03 | 20.64 | 20.56 | 45.50 | 62.78 | 29.03 | **11.50** | 25.00 | 23.06 | 51.52 | 28.47 |
| SNAPKV | 0.25 | 30.02 | 45.20 | 55.16 | 58.30 | **52.01** | 32.56 | 33.95 | 24.23 | 26.69 | 28.50 | 85.18 | 39.82 | 11.20 | 99.50 | **54.01** | 48.06 | 45.27 |
| | 0.5 | 28.98 | 41.20 | 45.34 | 56.93 | **49.53** | 29.89 | 32.12 | 23.59 | 25.78 | 33.00 | 86.08 | **40.48** | 9.55 | **99.50** | 53.21 | 47.29 | 43.90 |
| | 0.75 | 28.50 | 31.47 | 35.40 | 54.62 | 41.94 | **28.26** | 29.01 | 22.17 | 23.14 | 38.00 | 84.51 | **40.95** | 9.10 | 90.00 | **54.18** | 47.84 | 41.19 |
| | 0.9 | 24.15 | 20.79 | 22.81 | 44.52 | 23.38 | 20.51 | 25.14 | 19.70 | 20.07 | 34.50 | 81.21 | 38.33 | 6.50 | 53.50 | 51.06 | 48.96 | 33.45 |
| SINK | 0.25 | 28.29 | 43.53 | 37.19 | 52.35 | 42.17 | 32.69 | 31.92 | 23.97 | 26.33 | 30.50 | **91.81** | 38.57 | 8.44 | 76.00 | 47.02 | 47.34 | 41.13 |
| | 0.5 | 25.37 | 37.53 | 32.06 | 49.57 | 38.27 | 23.83 | 30.97 | 22.23 | 25.83 | 31.00 | **91.90** | 37.62 | 6.95 | 54.00 | 49.54 | 48.45 | 37.82 |
| | 0.75 | 23.27 | 24.47 | 25.44 | 42.06 | 27.26 | 19.82 | 28.87 | 20.82 | 23.80 | 32.00 | **91.52** | 35.66 | 7.00 | 33.50 | 50.90 | 50.51 | 33.56 |
| | 0.9 | 21.53 | 18.98 | 22.64 | 37.24 | 22.15 | 14.53 | 24.99 | 19.11 | 19.76 | 28.50 | 90.81 | 34.56 | 4.00 | 16.00 | 52.72 | **52.73** | 30.02 |
| CAP(OURS) | 0.25 | **31.53** | **47.74** | **56.95** | 57.31 | 51.90 | 32.91 | 34.61 | 24.65 | 26.78 | **68.50** | 90.21 | 38.80 | **13.65** | **100.00** | 53.47 | **48.52** | **48.60** |
| | 0.5 | 31.69 | 44.84 | 51.42 | **58.24** | 48.35 | 30.66 | **33.88** | 24.65 | 26.47 | **65.25** | 90.71 | 37.72 | 12.00 | 98.50 | 50.12 | **49.47** | **47.12** |
| | 0.75 | 29.73 | **39.93** | **43.71** | 53.19 | **46.49** | 24.66 | 29.32 | 23.71 | 25.19 | 30.75 | 90.83 | 33.44 | 9.00 | 66.50 | 44.47 | **50.91** | 40.11 |
| | 0.9 | 25.33 | 24.72 | 27.81 | 43.90 | 30.86 | 21.58 | 24.56 | 22.14 | 21.54 | 7.00 | 91.22 | 28.79 | 10.00 | 21.50 | 35.61 | 51.06 | 30.48 |

*Table 7.* Experimental results of Mistral0.3–7B on Longbench Benchmark.

| | C.R. | SD-QA | | | MD-QA | | | SUMM. | | | FSL | | | SYNTH. | | CODE | | AVG. |
|---|---|---|---|---|---|---|---|---|---|---|---|---|---|---|---|---|---|---|
| | | NQA | QSER | MF | HPQA | 2WQA | MQUE | GR | QMS | MN | TREC | TQA | SSUM | PC | PR | LCC | RBP | |
| MISTRAL0.3–7B | | 28.29 | 40.20 | 51.49 | 48.95 | 36.85 | 27.74 | 34.53 | 25.38 | 26.89 | 55.75 | 85.09 | 20.52 | 5.46 | 98.00 | 49.72 | 56.00 | 43.18 |
| EA | 0.25 | 26.90 | **39.23** | 50.98 | **49.12** | 36.57 | 26.35 | 33.97 | 25.03 | **26.87** | 55.75 | 86.42 | 21.22 | 4.09 | 96.50 | **51.34** | 55.30 | 42.85 |
| | 0.5 | 26.37 | 36.98 | 48.82 | **48.84** | 35.89 | 24.45 | **34.14** | 24.23 | 26.57 | 50.14 | 84.59 | 25.52 | 4.77 | 94.50 | **53.17** | 56.67 | 42.23 |
| | 0.75 | 25.10 | 32.36 | 42.76 | 46.55 | 38.98 | 23.15 | 32.04 | 22.96 | 26.04 | 54.75 | 85.40 | 39.88 | **5.13** | 76.00 | 54.17 | 57.53 | **41.42** |
| | 0.9 | **23.57** | 24.43 | 33.70 | 40.14 | 32.37 | 20.26 | 29.55 | 22.06 | 24.45 | 49.25 | 86.95 | 36.61 | **5.45** | 27.53 | 52.25 | 57.44 | **35.38** |
| KEYDIFF | 0.25 | 27.12 | 38.74 | **52.01** | 46.12 | **39.57** | **26.95** | 33.67 | 25.61 | 26.64 | 55.05 | 86.40 | 25.64 | 3.73 | **98.50** | 50.14 | 56.89 | 43.30 |
| | 0.5 | 26.09 | 34.49 | **49.54** | 45.05 | **36.52** | 25.03 | 32.22 | 24.69 | 25.79 | 52.00 | **88.11** | 35.84 | 3.85 | **95.00** | 46.75 | **59.53** | 42.53 |
| | 0.75 | 26.23 | 24.90 | 41.61 | 43.40 | 36.53 | 21.14 | 29.29 | 22.74 | 24.06 | 44.50 | **88.18** | **46.67** | 2.91 | **77.67** | 38.35 | **58.07** | 39.14 |
| | 0.9 | 17.77 | 15.72 | **34.68** | 34.51 | 28.94 | 16.30 | 26.10 | 21.00 | 21.17 | 36.00 | **87.14** | **44.81** | 3.17 | **42.50** | 30.05 | **57.59** | 32.34 |
| KNORM | 0.25 | 24.53 | 35.73 | 48.89 | 47.87 | 37.12 | 25.31 | 32.10 | 24.65 | 25.93 | 49.75 | **86.55** | 29.13 | 4.33 | 83.75 | 37.60 | 54.55 | 40.49 |
| | 0.5 | 19.98 | 28.78 | 41.90 | 46.08 | 31.24 | 21.81 | 29.99 | 23.24 | 24.64 | 41.00 | 86.73 | 36.52 | 3.71 | 66.50 | 32.76 | 54.10 | 36.81 |
| | 0.75 | 15.17 | 15.70 | 32.17 | 38.60 | 23.78 | 17.29 | 26.64 | 21.02 | 22.32 | 34.50 | 85.94 | 40.18 | 4.93 | 30.00 | 26.24 | 54.61 | 30.57 |
| | 0.9 | 12.72 | 6.03 | 24.84 | 29.42 | 27.77 | 12.67 | 24.00 | 19.00 | 19.89 | 26.75 | 85.80 | 38.61 | 4.00 | 10.50 | 23.08 | 55.14 | 26.26 |
| SNAPKV | 0.25 | 26.13 | 36.74 | 48.13 | 47.43 | 37.68 | 26.35 | 33.45 | 24.58 | 26.20 | 53.75 | 85.09 | 20.79 | 5.66 | 98.00 | 50.99 | 56.59 | 42.35 |
| | 0.5 | 23.71 | 32.12 | 46.55 | 47.59 | 37.10 | 23.31 | 31.49 | 23.66 | 25.33 | 50.50 | 85.89 | 21.02 | 4.82 | **96.50** | 52.66 | 57.05 | 41.21 |
| | 0.75 | 22.69 | 22.07 | 39.08 | 44.66 | 28.84 | 21.10 | 28.63 | 21.82 | 23.23 | 45.25 | 87.26 | 23.28 | 4.36 | **91.50** | 54.07 | 55.99 | 38.36 |
| | 0.9 | 18.34 | 13.63 | 29.69 | 33.02 | 22.60 | 18.66 | 25.10 | 20.41 | 20.64 | 38.25 | **87.68** | 27.01 | 4.50 | **53.00** | 52.03 | 54.72 | 32.45 |
| SINK | 0.25 | 24.80 | 36.89 | 35.52 | 45.02 | 30.96 | 23.68 | 33.43 | 23.87 | 25.94 | **61.50** | 77.22 | 19.21 | 2.82 | 77.00 | 51.21 | 54.99 | 39.00 |
| | 0.5 | 23.61 | 30.87 | 30.55 | 40.61 | 30.15 | 18.11 | 31.66 | 22.83 | 25.36 | **56.50** | 70.71 | 21.34 | 1.50 | 53.50 | 51.73 | 53.61 | 35.16 |
| | 0.75 | 20.09 | 19.51 | 26.10 | 39.21 | 26.51 | 16.79 | 28.84 | 21.44 | 22.61 | 42.00 | 62.55 | 26.84 | 2.07 | 32.50 | 51.62 | 53.24 | 30.74 |
| | 0.9 | 18.64 | 13.03 | 23.41 | 30.76 | 20.09 | 13.49 | 25.35 | 19.66 | 18.90 | 34.50 | 52.59 | 32.19 | 4.00 | 15.00 | 50.71 | 53.39 | 26.61 |
| CAP(OURS) | 0.25 | **28.40** | 38.72 | 50.11 | 48.40 | 37.68 | 26.46 | **34.49** | **25.74** | 26.74 | 52.25 | 85.28 | **30.40** | **5.85** | 96.50 | 51.29 | **57.09** | **43.46** |
| | 0.5 | **26.64** | **37.38** | 49.18 | 47.59 | 36.40 | **27.20** | 33.63 | **24.99** | **26.70** | 54.50 | 87.12 | **38.32** | **5.28** | 93.00 | 52.42 | 57.85 | **43.64** |
| | 0.75 | **26.30** | **32.82** | **43.42** | 45.91 | **39.73** | **23.17** | 30.85 | **23.56** | **26.09** | 41.00 | 86.04 | 44.80 | 4.29 | 59.50 | **58.35** | 57.54 | 40.21 |
| | 0.9 | 23.16 | **24.70** | 30.71 | **40.99** | **32.93** | 16.49 | 27.95 | 21.86 | **24.50** | 27.50 | 85.41 | 40.65 | 3.78 | 21.55 | **56.98** | 56.60 | 33.48 |

*Table 8.* Experimental results of Qwen3-4B on Longbench Benchmark.

| | C.R. | SD-QA | | | MD-QA | | | SUMM. | | | FSL | | | SYNTH. | | CODE | | AVG. |
|---|---|---|---|---|---|---|---|---|---|---|---|---|---|---|---|---|---|---|
| | | NQA | QSER | MF | HpQA | 2WQA | MQUE | GR | QMS | MN | TREC | TQA | SSUM | PC | PR | LCC | RBP | |
| QWEN3-4B | | 27.26 | 41.10 | 56.13 | 60.50 | 46.34 | 35.17 | 31.23 | 24.90 | 25.32 | 48.50 | 87.18 | 39.75 | 6.00 | 99.00 | 61.88 | 55.78 | 46.63 |
| EA | 0.25 | 26.01 | 38.90 | **55.49** | 61.52 | 44.06 | 34.20 | **31.77** | 24.36 | 25.11 | 51.50 | **88.01** | 39.82 | 5.50 | 98.00 | 62.96 | 56.23 | 46.47 |
| | 0.5 | **27.18** | 36.97 | 50.03 | **59.75** | 43.73 | 34.91 | **31.25** | 23.92 | **25.00** | 62.50 | **87.95** | 39.66 | 5.50 | 82.50 | 62.62 | 57.30 | 45.67 |
| | 0.75 | 24.21 | **33.10** | 41.01 | **54.27** | **40.61** | 27.38 | **29.67** | 22.39 | 24.14 | **66.00** | 86.98 | 39.73 | **7.00** | 26.75 | 57.27 | 58.74 | 39.95 |
| | 0.9 | **21.48** | **26.67** | 30.68 | **45.94** | 30.61 | 19.59 | 27.58 | 21.07 | 22.18 | 61.00 | 84.29 | **38.70** | 6.00 | 10.00 | 49.63 | 58.69 | **34.63** |
| KEYDIFF | 0.25 | 23.62 | 33.88 | 48.09 | 47.84 | 37.15 | 26.08 | 31.46 | 23.45 | **25.71** | 60.00 | 83.47 | 36.84 | **6.19** | 92.00 | 50.83 | 48.85 | 42.22 |
| | 0.5 | 21.95 | 25.78 | 38.22 | 36.43 | 28.91 | 17.54 | 26.66 | 22.27 | 21.59 | 51.50 | 81.87 | 35.60 | 4.12 | 79.50 | 32.09 | 43.74 | 35.49 |
| | 0.75 | 16.07 | 14.81 | 28.95 | 25.84 | 23.62 | 10.55 | 18.04 | 21.09 | 12.28 | 32.00 | 72.58 | 34.14 | 3.01 | 45.50 | 20.80 | 42.26 | 26.35 |
| | 0.9 | 11.28 | 9.97 | 18.48 | 14.79 | 25.41 | 7.09 | 11.83 | 19.49 | 7.72 | 10.50 | 67.14 | 27.59 | 4.27 | 13.50 | 14.09 | 45.72 | 19.30 |
| KNORM | 0.25 | 21.34 | 29.10 | 47.68 | 44.79 | 37.86 | 25.71 | 30.58 | 23.32 | 25.18 | 58.00 | 81.57 | 39.68 | 3.62 | 79.55 | 38.32 | 43.06 | 39.34 |
| | 0.5 | 17.45 | 16.22 | 36.63 | 33.76 | 32.22 | 17.60 | 21.20 | 22.23 | 16.94 | 38.50 | 80.35 | 38.78 | **6.91** | 58.33 | 24.55 | 41.33 | 31.44 |
| | 0.75 | 12.70 | 10.80 | 23.51 | 18.37 | 24.45 | 4.78 | 13.07 | 19.86 | 8.68 | 20.00 | 73.83 | 34.64 | 4.54 | 23.00 | 13.32 | 39.14 | 21.54 |
| | 0.9 | 7.40 | 9.95 | 17.19 | 11.26 | 19.03 | 4.38 | 7.80 | 18.79 | 5.01 | 4.00 | 67.17 | 29.64 | 4.90 | 4.50 | 12.50 | 41.28 | 16.55 |
| SNAPKV | 0.25 | 26.50 | 37.86 | 53.48 | 61.22 | **45.19** | **35.50** | 31.07 | **24.78** | 25.16 | 47.50 | 87.52 | 40.39 | 6.00 | **99.00** | 63.01 | 56.76 | 46.31 |
| | 0.5 | 27.09 | 33.06 | 43.38 | 58.28 | 41.74 | 32.09 | 29.92 | 23.32 | 23.70 | 45.50 | 87.52 | 40.16 | 4.50 | **99.00** | **63.87** | 58.54 | 44.48 |
| | 0.75 | 23.69 | 26.15 | 35.84 | 49.93 | 34.15 | 26.72 | 27.37 | 21.66 | 21.64 | 44.50 | 87.35 | **39.75** | 5.00 | **86.00** | **64.62** | 59.44 | 40.86 |
| | 0.9 | 19.66 | 17.46 | 26.48 | 39.52 | 28.60 | 21.06 | 23.19 | 19.26 | 17.62 | 28.00 | 87.19 | 38.09 | 5.50 | **39.92** | **61.72** | 60.27 | 33.35 |
| SINK | 0.25 | 23.18 | 36.95 | 36.48 | 56.18 | 36.22 | 27.19 | 29.03 | 23.28 | 24.58 | 41.00 | 83.94 | 39.37 | 2.50 | 76.00 | 60.37 | 54.19 | 40.65 |
| | 0.5 | 21.50 | 31.11 | 31.52 | 51.49 | 37.51 | 22.89 | 28.29 | 22.24 | 23.99 | 45.50 | 82.03 | 37.78 | 3.50 | 53.50 | 59.33 | 55.02 | 37.95 |
| | 0.75 | 20.25 | 22.29 | 25.83 | 43.50 | 35.68 | 17.87 | 26.78 | 20.88 | 21.40 | 42.50 | 78.08 | 36.81 | 4.00 | 32.50 | 58.49 | 55.53 | 33.90 |
| | 0.9 | 17.51 | 16.10 | 23.35 | 35.28 | 28.59 | 12.66 | 22.67 | 19.13 | 17.14 | 43.00 | 69.95 | 33.59 | 4.50 | 13.50 | 56.59 | 56.96 | 29.41 |
| CAP(OURS) | 0.25 | **26.76** | **39.59** | 55.33 | **62.05** | 45.16 | 34.63 | 31.57 | 24.72 | 25.35 | **67.50** | 87.35 | **40.55** | 4.50 | 98.00 | **63.04** | **59.70** | **47.86** |
| | 0.5 | 25.69 | **37.77** | **52.09** | 58.64 | **43.84** | **35.21** | 30.75 | **24.24** | 24.39 | 61.00 | 87.93 | **40.30** | 4.00 | 98.67 | 57.23 | **60.79** | **46.41** |
| | 0.75 | **25.30** | 31.09 | **42.60** | 53.39 | 36.58 | **31.52** | 29.20 | **23.74** | 22.53 | 52.50 | **87.85** | 38.93 | 6.00 | 81.25 | 46.25 | **61.64** | **41.90** |
| | 0.9 | 20.96 | 23.99 | **33.08** | 43.29 | **30.66** | **21.53** | 25.73 | 21.05 | 19.40 | 23.75 | **87.35** | 37.24 | **6.00** | 26.00 | 35.55 | **60.75** | 32.27 |

