# OpenReview forum: "Rethinking KV Cache Eviction via a Unified Information-Theoretic Objective"
_ICML.cc/2026/Conference — ICML 2026 regular_

### Official Review · Reviewer_ENMy · 2026-02-20

**Soundness:** 2
**Presentation:** 3
**Significance:** 3
**Originality:** 2
**Overall Recommendation:** 2
**Confidence:** 4

**Summary:**

The authors present CapKV (Capacity-Inspired KV-Cache Eviction), which reduces the kv-cache of transformer models by dropping kv-pairs with low relevance.

As it is common for kv-cache reduction methods, the paper is concerned in defining a relevant measure of importance.
The method uses an information theoretic approach and formulates the problem as finding an approximate kv-cache that maximizes, given a fixed budget, the relative information w.r.t. full kv-cache while being a good predictor for the attention output distribution.

Under several assumptions and approximations, their algorithm assigns each kv-pair a score, which consists of two contributions:
- an importance score $w_i$, which can be identified to measure how important the key is w.r.t. mean query
- a capacity score $u^⊤_i * A^{−1} * u_i$

The authors present several benchmarks, showing that CapKV outperforms several other compression methods such as Expected Attention or KeyDiff.

**Compliance With Llm Reviewing Policy:**

Affirmed.

**Final Justification:**

After a carefully reading of the answers I decide to keep my score because I still see the theoretical framework flawed.

**Key Questions For Authors:**

1. Can you elaborate how your account for ROPE in your method when modeling the query statistics?
2. Given that $W_O$ is a learned transformation, what supports the approximation $u_i \approx v_i$?
3. You introduce a softmax-like exponential weighting. This seems to contradict the main assumption of the paper. Can you elaborate on this?

**Limitations:**

yes

**Strengths And Weaknesses:**

## Strengths
* Viewing KV-cache eviction as a capacity-maximization problem is a novel and interesting approach.
* The method is computationally efficient and can be run fast with negligible overhead.
* Existing approaches, such as **Knorm**, can be interpreted as lower-order approximations of this information-theoretic framework.
* The capacity score ($u_i^\top A^{-1} u_i$) is potentially modular and could be integrated into other compression methods.

## Weaknesses
While the empirical evaluation is decent, the technical soundness of the paper's core theoretical claims is flawed. The authors make a variety of strong assumptions and simplifications that appear designed to arrive at an easily implementable algorithm, and there are several inconsistencies in the proofs:

* Assumption 3.1 introduces a linear attention mechanism to derive the importance score. While the authors acknowledge that this is not intended to faithfully model the full nonlinear dynamics of Transformers, it remains unclear if the insights from the linearized version translate to standard attention.
* Queries are modeled to follow a Gaussian distribution. This fails to account for **RoPE (Rotary Positional Embedding)** scaling; because the distribution of a future query is position-dependent, a generated sequence $[q_{t+1}, \dots, q_{T}]$ is more likely to follow a mixture of Gaussians.
* Lemma 3.3 relies on a first-order Taylor expansion that may break down for highly novel or highly relevant tokens.
* The output projection $W_O$ is dropped to arrive at the final result, which is only valid if $W_O$ is an isometry. It is not apparent how the removal of $W_O$ affects the validity of the scoring.
* To derive Equation 6, the authors assume isotropic query/noise distributions and ignore value-channel effects. These assumptions are not discussed in detail and may result in an oversimplified importance score.
* The core theorem (Theorem 3.2) relies on a linear attention model (Assumption 3.1). However, Equation 8 reintroduces a softmax-like exponential weighting ($w_i = \exp(k_i^\top \mu_q / \tau)$), which implicitly non-linearizes the algorithm and contradicts the theorem's foundational assumptions.
* The algorithm resembles the use of statistical leverage scores to maximize the spanned volume of the cache. The authors should compare their method to **"Compactor: Calibrated Query-Agnostic KV Cache Compression with Approximate Leverage Scores."**
* Several other KV-cache compression methods exist that outperform the selected comparison methods (e.g., **KVzip**). These should be added to benchmark the method against the current SOTA.

---

> ### Author Rebuttal · Authors · 2026-03-31
>
> ## 1. Linear-Gaussian Model validation
> We clarify that Assumption 3.1 is an analytical surrogate for tractability, not a claim of exact modeling. To justify its utility, we provide two-fold evidence on Qwen3-8B:
>
> ### Output Cosine Similarity
>
> To assess the validity of the linear-Gaussian approximation, we computed the cosine similarity between the outputs of the linear-Gaussian model ($Y_{CapKV}$) and the real attention mechanism ($Y_{full}$) on the Qwen3-8B model across different layers. The results showed that cosine similarities ranged from 0.6 to 0.9, indicating that the directionality of the two outputs is highly consistent. While magnitudes of $Y_{CapKV}$ and $Y_{full}$ differ due to the absence of Softmax normalization, the preservation of relevant directions is crucial for cache eviction.
>
> |Dataset|2wikimqa|qasper|trec|
> |-|-|-|-|
> |Layer2|0.60±0.0498|0.69±0.0380|0.71±0.0227|
> |Layer15|0.91±0.0247|0.91±0.0145|0.91±0.0121|
> |Layer20|0.86±0.0134|0.85±0.0098|0.88±0.0073|
> |Layer35|0.66±0.0486|0.74±0.0308|0.73±0.0298|
>
> ### Query and Noise Independence Validation
>
> We also evaluated the independence of the noise term $\epsilon$ from the query vector $q$ by measuring Pearson’s correlation coefficient ($\text{Corr}(\epsilon, q)$) across layers and datasets. Correlations were close to zero (below 0.012), supporting our assumption that $\epsilon$ does not contain significant linear information about the query.
>
> |Layer|2wikimqa|qasper|trec|
> |-|-|-|-|
> |Layer2|0.0027±0.0035|0.0029±0.0035|0.0014±0.0033|
> |Layer15|0.0117±0.0031|0.0081±0.0029|0.0115±0.0024|
> |Layer20|\-0.0026±0.0035|\-0.0021±0.0030|0.0056±0.0025|
> |Layer35|0.0003±0.0064|0.0049±0.0060|0.0017±0.0032|
>
>
> ### Theoretical Validity and Maximum Entropy Principle
> Theoretically, the Gaussian distribution is the most robust and minimally restrictive model under the maximum entropy principle when only first- and second-order moments are known. Consequently, even if actual noise is non-Gaussian (e.g., heavy-tailed), the log-determinant remains the optimal capacity measure for linear prediction. This is empirically validated by the strong Spearman correlation ($\rho \approx 0.8$) between our capacity proxies and downstream performance.
> ## Queries distribution
>
> Regarding the query distribution post-RoPE, we follow common analytical assumptions  where empirical evidence supports the Gaussianity of queries in Transformer layers[1][2]. This assumption enables a tractable information-theoretic bottleneck analysis without sacrificing the method's practical ranking capability.
>
> ## Ablation of Output Projection
> We tested the impact of incorporating $W_O$ ($u_i = W_O v_i$) on a LongBench subset. Results show that while including $W_O$ provides marginal gains, it dramatically increases complexity from $O(Nd^2 + d^3)$ to $O(N(dh)^2 + (dh)^3)$. Our approximation is a principled trade-off for inference efficiency.
>
> |CapKV|2WQA|MN|MF|PR|TREC|Avg.|
> |-|-|-|-|-|-|-|
> |C.R.|||||||
> |0.50|44.40|24.59|51.11|98.42|66.50|57.00|
> |0.75|38.44|23.12|44.16|96.96|60.50|52.64|
> |with_wo|||||||
> |0.50|43.76|24.76|51.88|98.02|67.50|57.18|
> |0.75|39.29|23.35|44.18|97.54|60.00|52.87|
>
> ## $\exp$ Coefficient
> We appreciate the reviewer for pointing out the issue regarding Assumption 3.1 and Equation (8). While there is a mathematical difference, this is an intentional decoupling between the theoretical derivation and the algorithmic instantiation.
>
> Assumption 3.1 enables the derivation of the mutual information objective in Theorem 3.2, which characterizes KV cache capacity. Equation (8) introduces exponential weighting $w_i = \exp(k_i^\top \mu_q / \tau)$ to inject an importance prior for each KV pair, without changing the linear nature of the channel. As shown in Table 2, setting $\tau=5$ significantly improves performance compared to $\tau=0$. This demonstrates that combining theoretical capacity (second-order) with relevance prior (first-order) is key to CapKV’s success.
>
> ## Additional Baseline
> As reviewer requested, we benchmarked CapKV against Compactor and KVZip using Qwen3-8B on Longbench:
> |method/C.R.|compactor|kvzip|CapKV|
> |-|-|-|-|
> |0.25|48.54|47.69|49.51|
> |0.5|46.98|48.08|48.15|
> |0.75|42.70|48.20|44.88|
> |0.9|33.35|39.60|36.57|
>
> We will provide the detailed results in the appendix. As the result show, While also using leverage scores, Compactor in lower average performance across all compression ratios. KVZip shows strength at high compression ratios but relies on a "repeat prompt" strategy to evaluate token importance. This makes it ~2x slower than representation-based methods, which maintains a negligible $O(Nd^2)$ overhead.
>
> We hope the above explanation clarifies your concerns, and I would be happy to further discuss how to improve the content of the paper.
>
> [1] Alessio Devoto. Expected Attention: KV Cache Compression by Estimating Attention from Future Queries Distribution, Oct 2025
>
> [2] James Liu. Training-Free Activation Sparsity in Large Language Models, Aug 2024

---

> > ### Author Rebuttal · Reviewer_ENMy · 2026-04-02
> >
> > Thanks a lot for the detailed answer!
> >
> > In general, both the cache eviction algorithm itself, as well as the experimental results are convincing and novel.
> >
> > I'm still observing a discrepancy in presentation: The authors start with a precise mathematical premise (use an information-theoretic approach). To obtain the final cache eviction formula, a strongly simplified and somewhat artifical toy model is used.
> > In my opinion, this is an indicator that the initial premise, while being useful to come up with an interesting result, is not a natural narrative/premise.
> >
> > To me, a more suitable fit would be a recognition of the final formula as consisting of a generic token importance score, modified by the capacity score. A mathematical anlysis/motivation of the two controbutions seems then to be an interesting and novel exercise.

---

> > > ### Author Response · Authors · 2026-04-03
> > >
> > > We thank the reviewer for the insightful comments regarding the presentation and narrative of our work. Due to text Length limitations in the previous rebuttal, we were unable to fully clarify the role of the exp weighting. To address this, we provide additional ablation results comparing CapKV-Linear (i.e., without exp weighting) with existing methods.
> > >
> > > We report the average performance on the 16 tasks of LongBench using Qwen3-8B below (full results will be included in the appendix):
> > >
> > > | C.R.\Method | Compactor | EA    | KeyDiff | Knorm | SnapKV | CapKV-Linear | CapKV-Exp |
> > > |------------|----------|------|--------|------|--------|-------------|----------|
> > > | 0.25       | 48.54    | 47.85 | 44.03   | 43.48 | 46.97  | 49.23        | 49.51     |
> > > | 0.5        | 46.98    | 47.12 | 39.01   | 35.72 | 45.36  | 47.45        | 48.15     |
> > > | 0.75       | 42.70    | 42.62 | 29.10   | 24.79 | 42.06  | 41.25        | 44.88     |
> > > | 0.9        | 33.35    | 36.35 | 20.95   | 20.09 | 35.28  | 32.34        | 36.57     |
> > >
> > > The CapKV-Linear variant, which directly corresponds to the original capacity objective (Theorem 3.2), already achieves strong performance. In particular, under moderate compression ratios (0.25, 0.5), it outperforms existing heuristic methods and is competitive with strong baselines. Even under aggressive compression, it consistently surpasses methods such as KeyDiff and KNorm. This indicates that the capacity-aware term alone provides a strong and theoretically grounded signal for KV selection.
> > >
> > > Both CapKV-Linear and CapKV-Exp incorporate query-dependent weighting, but differ in how they separate tokens by relevance. CapKV-Linear applies a relatively smooth (linear) weighting, whereas CapKV-Exp introduces a nonlinear amplification that increases the contrast between highly relevant and less relevant tokens.
> > >
> > > Under aggressive compression, eviction becomes a hard selection problem, where only a very small subset of tokens can be retained. In this regime, linear weighting may produce many tokens with similar scores, leading to ambiguous ranking and suboptimal selection. The exponential formulation enlarges score gaps, resulting in clearer separation and more reliable top-K selection. This explains the larger gains of CapKV-Exp in high-compression settings.
> > >
> > > Importantly, the exp weighting does not contradict the theoretical framework. Instead, it can be viewed as a practical refinement that preserves the same query-relevance component while improving selection robustness under strict budget constraints.
> > >
> > > Regarding the concern about the presentation, we fully agree that the current narrative can be improved. In the final version, we will revise the method section to explicitly present the importance–capacity decomposition first, and position the information-theoretic analysis as a principled lens that explains this structure.
> > >
> > > We sincerely thank the reviewer for this constructive suggestion, which we believe will significantly improve the clarity of the paper. We hope the additional analysis addresses your concerns and would greatly appreciate your reconsideration of the rating.

---

### Official Review · Reviewer_L3Ld · 2026-02-28

**Soundness:** 2
**Presentation:** 2
**Significance:** 2
**Originality:** 3
**Overall Recommendation:** 4
**Confidence:** 5

**Summary:**

This paper offers a novel perspective on KV cache eviction algorithms by framing the selection of KV representatives through the lens of the information bottleneck principle. It derives a formula for measuring information capacity and uses it as a unified framework for viewing various KV eviction methods. Leveraging this formula, the authors propose a method named **CapKV**, which performs KV cache pruning by approximately maximizing this information capacity. The authors then conduct extensive experiments to demonstrate the effectiveness of CapKV.

**Compliance With Llm Reviewing Policy:**

Affirmed.

**Final Justification:**

I have carefully re‑read all of the authors’ responses to my comments and, through our discussion, have gained a deeper understanding of this work. On the whole, I continue to hold my initial view and feel even more confident in my original score.

The paper presents extensive experiments and undoubtedly proposes an effective attention‑sparsification algorithm named CapKV. Theoretically, despite some remaining doubts, the framework is sufficient to justify the design of CapKV itself. In my assessment, what prevents this paper from receiving a higher score is that the proposed theory lacks a broader generality.

Therefore, my final decision is to maintain my initial rating.

**Key Questions For Authors:**

Please refer to the points raised in the **Weaknesses** section.

**Limitations:**

Yes

**Strengths And Weaknesses:**

Overall, given the well-organized experiments, I believe this paper presents a competent KV eviction method. However, I still have reservations regarding the information-theoretic discussions. If the authors can address these concerns, I would be willing to increase my score.

## Strengths

- The information bottleneck perspective proposed for KV eviction is intriguing.
- Through detailed theoretical derivation, the paper obtains a metric for evaluating KV eviction methods.
- The paper organizes comprehensive experiments, and the proposed CAPKV method demonstrates strong performance.

## Weaknesses

- The caption text in **Figure 1** does not display correctly on my computer, preventing me from understanding its content. A similar, though less severe, issue occurs in **Figure 2**. I suspect this is related to font usage. Please use the same font as in the main text whenever possible.
- **Figure 3** seems intended to show the correlation between capacity and performance. A concerning point is that the correlation appears to stem primarily from the performance of the *same method* under different Compression Ratios (C.R.). The relationship between capacity and performance among *different methods* at the *same C.R.* seems less evident. This directly weakens the support this experiment provides for the claimed correlation.
- Although the paper provides detailed theoretical derivations, it employs numerous approximations throughout the process. For readers who are not experts in the relevant fields, this somewhat undermines its persuasiveness. Beyond the explicitly stated assumption of a **linear-Gaussian surrogate model**, at least the following two approximations warrant further scrutiny:
 - Assuming the query distribution is Gaussian and that noise is independent of the query. The text suggests this is a concomitant assumption of the information bottleneck viewpoint, but I did not find a discussion on the reliability and rigor of this assumption. The lack of discussion on this point, combined with the second weakness, directly shakes the value and persuasiveness of the proposed information-theoretic evaluation perspective.
 - The multiple approximations used in the design of the **CapKV algorithm** due to computational complexity and other considerations. I understand that directly solving the original optimization problem may be impractical, and its impact might be difficult to analyze. However, I believe it might be possible to include a method that directly solves the original problem (as an **oracle**) for comparison in the experiments. This would significantly enhance the credibility of both the proposed method and the experimental setup overall.

---

> ### Author Rebuttal · Authors · 2026-03-31
>
> ## Display Issues
> We sincerely apologize for the display issues in Figures 1 and 2. We will correct the font usage in the revised version to ensure the figures are readable across different systems.
>
> ## Capacity-Score Spearman Correlation
> To isolate compression ratio effects, we report Spearman correlations between capacity metrics and task performance:
>
> Spearman Correlation (CR = 0.9):
> - 2WQA:
>   - K Capacity: -0.14 (p-value = 0.7872)
>   - U Capacity: 0.42 (p-value = 0.3965)
>   - KU Capacity: 0.88 (p-value = 0.0188)
>
> - PR:
>   - K Capacity: 0.08 (p-value = 0.8717)
>   - U Capacity: 0.54 (p-value = 0.2657)
>   - KU Capacity: 0.82 (p-value = 0.0415)
>
> - qasper:
>   - K Capacity: 0.02 (p-value = 0.9572)
>   - U Capacity: 0.54 (p-value = 0.2657)
>   - KU Capacity: 0.82 (p-value = 0.0416)
>
> These results suggest that when the compression ratio's influence is controlled, the combined capacity metric (KU Capacity) shows a significantly stronger correlation with downstream task performance compared to individual capacity metrics. Additional CR settings will be provided in the appendix.
>
> ## Linear-Gaussian Model vs. Real Attention Model
> We appreciate the reviewer’s thoughtful concerns about the linear-Gaussian model. While this surrogate model simplifies the attention mechanism, we have conducted extensive empirical analyses to demonstrate its utility for Key-Value (KV) cache eviction tasks. Below, we address how this model compares with the real attention mechanism.
>
> ### Output Cosine Similarity
> To evaluate the linear-Gaussian approximation's validity, we computed the cosine similarity between the outputs of the linear-Gaussian model ($Y_{CapKV}$) and the real attention mechanism ($Y_{full}$) on the Qwen3-8B model across multiple layers. The results show that cosine similarities range from 0.6 to 0.9, suggesting that the directionality of the outputs is highly consistent. While magnitudes of $Y_{CapKV}$ and $Y_{full}$ differ due to the absence of Softmax normalization, the preservation of relevant directional information is critical for cache eviction.
>
> |Dataset|2wikimqa|qasper|trec|
> |-|-|-|-|
> |Layer2|0.60±0.0498|0.69±0.0380|0.71±0.0227|
> |Layer15|0.91±0.0247|0.91±0.0145|0.91±0.0121|
> |Layer20|0.86±0.0134|0.85±0.0098|0.88±0.0073|
> |Layer35|0.66±0.0486|0.74±0.0308|0.73±0.0298|
>
> ### Query and Noise Independence Validation
>
> We also assessed the independence of the noise term $\epsilon$ from the query vector $q$ by measuring the Pearson correlation coefficient ($\text{Corr}(\epsilon, q)$) across layers and datasets. The correlations were close to zero (below 0.012), reinforcing our assumption that $\epsilon$ does not carry significant linear information about the query, thereby justifying its treatment as independent noise in our theoretical framework.
>
> |Layer|2wikimqa|qasper|trec|
> |-|-|-|-|
> |Layer2|0.0027±0.0035|0.0029±0.0035|0.0014±0.0033|
> |Layer15|0.0117±0.0031|0.0081±0.0029|0.0115±0.0024|
> |Layer20|\-0.0026±0.0035|\-0.0021±0.0030|0.0056±0.0025|
> |Layer35|0.0003±0.0064|0.0049±0.0060|0.0017±0.0032|
>
> ### Theoretical Validity
> The Gaussian distribution is considered the most robust and minimally restrictive model when only the first- and second-order moments (mean and covariance) of noise are known, according to the maximum entropy principle. Even when the actual noise distribution may exhibit non-Gaussian features (such as heavy tails), the log-determinant metric derived from the Gaussian assumption remains the optimal capacity measure for linear prediction scenarios. This is corroborated by the high Spearman correlation ($\rho \approx 0.8$) between our capacity predictions and downstream performance, further validating the effectiveness of our information-theoretic framework.
>
> ## Oracle Model
> In response to the reviewer’s suggestion about comparing the performance of the approximated CapKV model with a direct oracle solution. We have conducted additional ablation experiments using Qwen3-8B on a subset of tasks from the Longbench dataset. Below are the preliminary results:
>
> |CapKV|2WQA|MN|MF|PR|TREC|Avg.|
> |-|-|-|-|-|-|-|
> |C.R.|||||||
> |0.50|44.40|24.59|51.11|98.42|66.50|57.00|
> |0.75|38.44|23.12|44.16|96.96|60.50|52.64|
> |with_wo|||||||
> |0.50|43.76|24.76|51.88|98.02|67.50|57.18|
> |0.75|39.29|23.35|44.18|97.54|60.00|52.87|
> |with_conv|||||||
> |0.50|47.26|23.98|50.29|97.06|64.00|56.52|
> |0.75|39.71|21.84|43.63|94.92|46.00|49.22|
>
> These results indicate that using $u_i = W_O v_i$ offers a slight performance improvement, the computational cost increases dramatically, changing the complexity from $O(Nd^2 + d^3)$ to $O(N(d \cdot h)^2 + (d \cdot h)^3)$. Therefore, the trade-off in performance due to CapKV’s approximation is justified.
>
> Furthermore, using Conv-based query estimations leads to slightly decreased performance, likely due to the instability introduced by estimating queries using historical data (compared to the more stable mean-based approach).
>
> We hope the response above addresses your concerns and kindly ask you to raise your score.

---

> > ### Author Rebuttal · Reviewer_L3Ld · 2026-04-03
> >
> > I acknowledge that the authors have responded to my specific concerns and largely addressed my reservations. However, the responses, while clarifying the experimental aspects, do not sufficiently elevate the theoretical contribution to warrant a higher score.
> >
> > My initial concerns centered on the dual insufficiency of both experimental validation and theoretical grounding. The authors have mitigated the experimental limitations by providing additional data—such as the fixed-compression-ratio correlation analysis and the cosine-similarity measurements—and have used downstream performance evidence to support the validity of the theoretical derivation. While this line of reasoning is not fundamentally flawed, it does not fully resolve the deeper issue of theoretical self-consistency. The theoretical framework remains constrained by strong simplifications (e.g., the linear-Gaussian assumption, the approximation \(u_i \approx v_i\), and the disregard of RoPE and softmax nonlinearities). As a result, the theory appears more as a tailored justification for CapKV itself rather than a universally applicable evaluation criterion for all attention‑sparsification algorithms.
> >
> > Consequently, I maintain my original rating.

---

> > > ### Author Response · Authors · 2026-04-04
> > >
> > > We thank the reviewer for the careful follow-up and for acknowledging the improvements in the experimental validation.
> > >
> > > Regarding the theoretical aspect, we agree that the linear-Gaussian formulation relies on strong simplifications (e.g., ignoring softmax nonlinearity, RoPE, and using approximations such as $u_i \approx v_i$). Our intention is not to provide a fully self-consistent model of attention, but rather a tractable abstraction that highlights the dominant structural factors in KV cache eviction.
> > >
> > > In this sense, the framework should be interpreted as a first-order analytical lens, rather than a universally exact evaluation criterion. The simplifying assumptions enable a closed-form objective that captures interactions between key diversity, value directions, and query statistics, which we find to align well with empirical behavior.
> > >
> > > We also clarify that the framework is not meant to be specific to CapKV; instead, CapKV is one practical instantiation, while other methods can be viewed as emphasizing different components of the same objective under additional assumptions. We will revise the paper to better reflect this scope and avoid overstatement.
> > >
> > > We appreciate the reviewer’s insightful comments, which help us better position the theoretical contribution. We also sincerely appreciate your maintaining a positive recommendation of our work.

---

### Official Review · Reviewer_LRSo · 2026-03-11

**Soundness:** 4
**Presentation:** 3
**Significance:** 3
**Originality:** 3
**Overall Recommendation:** 4
**Confidence:** 3

**Summary:**

This paper formulate Key-Value (KV) cache eviction in LLM inference through the lens of the Information Bottleneck principle. By introducing a linear-Gaussian proxy model for attention, the authors derived a mutual information objective that unifies different heuristic eviction strategies under a single capacity-maximization framework. Building on this theoretical insight, they propose CapKV, an eviction algorithm that directly targets information preservation using statistical leverage scores derived from a capacity matrix. Extensive experiments on long-context benchmarks demonstrate that CapKV consistently outperforms existing heuristic methods across multiple model architectures and compression ratios.

**Compliance With Llm Reviewing Policy:**

Affirmed.

**Key Questions For Authors:**

1. The theoretical model incorporates all unmodeled factors such as softmax nonlinearity and multi-head interaction into a zero-mean Gaussian noise term independent of the query. Could the authors provide validation of the effectiveness of this assumption in actual Transformer attention, such as a quantitative analysis of the distribution characteristics of this noise term, its correlation with the query, and the impact of approximation errors on the final mutual information objective? The answer to this question will directly affect our evaluation of the rigor and technical completeness of the theoretical framework; the lack of effective validation will significantly weaken the credibility of the paper's core claims.
2. Existing experiments have only been validated under a maximum context length of 64k, while the core application scenario for KV cache eviction is extreme long-context inference with 128k and above. In such cases, the memory bottleneck becomes more prominent, and the redundancy patterns of KV may also change. Could the authors supplement performance and efficiency experiments in ultra-long context scenarios of 128k and above to verify the generalization ability of CAPKV in its core target scenarios? The answer to this question will directly affect our evaluation of the practical value of the method.
3. The paper uses existing pure eviction methods as baselines, but in actual industrial deployments, KV cache optimization usually adopts a hybrid strategy combining eviction, quantization, and merging. Could the authors verify the compatibility of CAPKV with other KV cache optimization methods such as per-channel value-cache quantization and whether CAPKV can still maintain its performance advantage over other eviction strategies in hybrid optimization schemes? The answer to this question will verify the practical application value of the method.

**Strengths And Weaknesses:**

- Strengths: 1) Theoretical Unification: By employing a linear-Gaussian surrogate model, it derives a closed-form mutual information objective that elegantly unifies disparate existing heuristic strategies under a single capacity-maximization framework. 2) Strong Empirical Correlation: The authors successfully bridge theory and practice by demonstrating a robust, statistically significant positive correlation between their proposed retained capacity proxies and actual downstream task performance across multiple datasets. 3) Solid Generalization: CapKV consistently outperforms strong baselines across diverse model architectures and complex reasoning benchmarks (e.g., LongBench, AIME25).
- Weakness: 1) The paper lacks quantitative validation regarding the actual distribution of this noise term in real Transformers and how the approximation error impacts the final mutual information objective, which weakens the rigor of the theoretical claims. 2) Lack of Extreme Long-Context Evaluation: The redundancy patterns and sparsity of attention may shift fundamentally at long-context, leaving CapKV's generalization ability in its most critical target scenario unverified.

---

> ### Author Rebuttal · Authors · 2026-03-31
>
> ## 1. Linear-Gaussian Model validation
> We thank the reviewer for highlighting concerns regarding the linear-Gaussian approximation. While this surrogate model simplifies the complex attention mechanism, we have conducted extensive empirical analyses to validate its utility for Key-Value (KV) cache eviction tasks. We hope to clarify how this model performs relative to the real attention mechanism.
>
> ### Output Cosine Similarity
>
> To assess the validity of the linear-Gaussian approximation, we computed the cosine similarity between the outputs of the linear-Gaussian model ($Y_{CapKV}$) and the real attention mechanism ($Y_{full}$) on the Qwen3-8B model across different layers. The results showed that cosine similarities ranged from 0.6 to 0.9, indicating that the directionality of the two outputs is highly consistent. While magnitudes of $Y_{CapKV}$ and $Y_{full}$ differ due to the absence of Softmax normalization, the preservation of relevant directions is crucial for cache eviction.
>
> |Dataset|2wikimqa|qasper|trec|
> |-|-|-|-|
> |Layer2|0.60±0.0498|0.69±0.0380|0.71±0.0227|
> |Layer15|0.91±0.0247|0.91±0.0145|0.91±0.0121|
> |Layer20|0.86±0.0134|0.85±0.0098|0.88±0.0073|
> |Layer35|0.66±0.0486|0.74±0.0308|0.73±0.0298|
>
> ### Query and Noise Independence Validation
>
> We also evaluated the independence of the noise term $\epsilon$ from the query vector $q$ by measuring Pearson’s correlation coefficient ($\text{Corr}(\epsilon, q)$) across layers and datasets. Correlations were close to zero (below 0.012), supporting our assumption that $\epsilon$ does not contain significant linear information about the query.
>
> |Layer|2wikimqa|qasper|trec|
> |-|-|-|-|
> |Layer2|0.0027±0.0035|0.0029±0.0035|0.0014±0.0033|
> |Layer15|0.0117±0.0031|0.0081±0.0029|0.0115±0.0024|
> |Layer20|\-0.0026±0.0035|\-0.0021±0.0030|0.0056±0.0025|
> |Layer35|0.0003±0.0064|0.0049±0.0060|0.0017±0.0032|
>
>
> ### Theoretical Validity and Maximum Entropy Principle
> From a theoretical standpoint, it is well known that the Gaussian distribution is the most robust and minimally restrictive model when only the first- and second-order moments (mean and covariance) of noise are known, according to the maximum entropy principle. Even when the actual noise distribution may exhibit non-Gaussian features (such as heavy tails), the log-determinant metric derived from this Gaussian assumption remains the optimal capacity measure for linear prediction scenarios. This is corroborated by the high Spearman correlation ($\rho \approx 0.8$) between our capacity predictions and downstream performance, which further validates the effectiveness of our information-theoretic framework.
>
> ## 2. Experiments on Extreme Long-Context Inference
> Regarding the reviewer’s concern about performance in extreme long-context settings, we have extended our evaluation by conducting experiments on the LongbenchV2 dataset. We specifically selected data with context lengths ranging from 32k to 128k, totaling 118 samples, and performed initial tests using the Qwen3-8B model with yarn extrapolation.
> The results is shown in table, demonstrate that CapKV outperforms other baseline methods at higher compression ratios (0.75–0.9):
>
> |Methods/C.R.|EA|keydiff|knorm|snapkv|CapKV|
> |-|-|-|-|-|-|
> |No Compression Qwen3-8B(44.5)|
> |0.5|45.4|29.4|31.9|46.2|43.7|
> |0.75|45.4|31.1|23.5|38.7|47.9|
> |0.9|43.7|22.7|16.8|42.9|44.8|
>
> ## 3. Compatibility with Quantization Techniques
> Regarding the reviewer’s query about the compatibility of CapKV with existing quantization techniques, we would like to highlight that CapKV is a model-agnostic structural compression technique, orthogonal to representation compression methods. Thus, it is fully compatible with existing quantization techniques. To assess this, we tested CapKV on a subset of tasks from the Longbench dataset, using the HQQ quantization method [1] at a 4-bit level.
>
> The results is shown in table. The HQQ quantization results highlight that CapKV consistently outperforms other baselines at both compression ratios. We will provide further detailed experimental results in the appendix.
> |method/CR|Tasks|||||||
> |-|-|-|-|-|-|-|-|
> |EA|2WQA|Lcc|MN|MF|PR|TREC|Avg.|
> |0.50|46.78|65.41|24.94|51.11|87.46|64.00|55.14|
> |0.75|39.27|59.30|23.70|39.60|49.74|70.00|42.32|
> |keydiff||||||||
> |0.50|33.86|46.51|21.93|39.86|86.33|55.00|45.70|
> |0.75|27.49|27.38|14.22|29.26|45.58|39.00|28.79|
> |Knorm||||||||
> |0.50|28.61|36.13|21.26|39.55|78.46|50.50|40.80|
> |0.75|18.25|16.83|14.73|27.76|20.00|25.75|19.51|
> |Snapkv||||||||
> |0.50|45.84|68.10|23.38|45.60|90.71|40.50|54.73|
> |0.75|38.90|67.73|21.10|35.60|88.09|38.50|50.28|
> |CapKV||||||||
> |0.50|44.54|59.47|24.47|51.02|97.85|66.50|55.47|
> |0.75|39.27|51.51|23.21|44.72|97.58|60.50|51.26|
>
> In light of the response above, we hope it can address your concerns and appreciate your suggestions. Thank you for your feedback and we kindly ask you to re-evaluate our submissions.
>
> [1] Hicham Badri and Appu Shaji.Half-quadratic quantization of large machine learning models, November 2023.

---

> > ### Author Rebuttal · Reviewer_LRSo · 2026-04-04
> >
> > Thank the authors for the rebuttal. I'll remain the positive score.

---

> > > ### Author Response · Authors · 2026-04-04
> > >
> > > We sincerely thank the reviewer for the positive evaluation and for acknowledging the improvements in our rebuttal. We appreciate the thoughtful feedback provided, and we are glad that our clarifications addressed the concerns raised. We value the reviewer’s support and constructive comments, which will guide us in further refining the paper for the final version.

---

### Official Review · Reviewer_NaPt · 2026-03-13

**Soundness:** 3
**Presentation:** 3
**Significance:** 3
**Originality:** 3
**Overall Recommendation:** 4
**Confidence:** 2

**Summary:**

This paper proposes an information-theoretic framework for KV cache eviction in LLM inference. By adopting a linear-Gaussian surrogate model of the attention mechanism, the authors derive a closed-form mutual information objective (a log-determinant expression) that characterizes the information capacity of a retained KV cache subset. They show that existing heuristic eviction methods (SnapKV, H2O, KeyDiff, Knorm, etc.) can be interpreted as approximating different components of this unified objective. Building on this analysis, they propose CapKV, which uses statistical leverage scores from a capacity matrix to greedily select which cache entries to retain. Experiments on LongBench (16 tasks) across multiple models (Qwen3-8B/14B, Llama3.1-8B, Mistral-7B) and a reasoning benchmark (AIME25 with Nemotron-7B) show consistent improvements over baselines.

**Compliance With Llm Reviewing Policy:**

Affirmed.

**Final Justification:**

My concerns are addressed and I will value the paper as the rating of 4.

**Key Questions For Authors:**

- How does the correlation between the linear-Gaussian surrogate's capacity prediction and actual mutual information (estimated empirically from attention distributions) vary across layers and sequence lengths? This would help calibrate how much to trust the theoretical framework.
- Why not estimate the query covariance Lambda_Q (even a diagonal approximation) instead of only using the mean? The theory suggests it matters, and the overhead would be minimal.

**Strengths And Weaknesses:**

## Strengths

- The unifying perspective is the real contribution here. Showing that attention-based methods (SnapKV, H2O) and structure-based methods (KeyDiff, Knorm) are approximating different terms of the same log-determinant objective is a genuinely useful insight. The analysis in Appendix B, where each method is mapped to a specific simplification of the capacity expression, is well-done and makes the relationships concrete rather than hand-wavy.

- The empirical validation of the theory is convincing. The capacity-performance correlation analysis (Spearman rho 0.73-0.87 across four datasets and three capacity proxies) provides strong evidence that the proposed information capacity metric actually tracks what matters for downstream performance. This is not just a theoretical exercise — the metric has predictive value.

- CapKV is practical. The O(Nd^2 + d^3) complexity is reasonable, the runtime plots (Figure 5) show it's in the same ballpark as existing methods, and the single hyperparameter tau is not overly sensitive (tau=5 works well across the board per Table 2). The method doesn't require any model modification or retraining.



## Weaknesses

- The linear-Gaussian assumption (Assumption 3.1) is doing a lot of heavy lifting. The entire theoretical framework rests on modeling attention as Y = U_C K_C q + epsilon, which strips away the softmax nonlinearity — arguably the defining feature of attention. The paper acknowledges this is a "surrogate" but doesn't quantify how much the approximation degrades as contexts get longer or attention patterns become more peaked. The correlation analysis is indirect evidence, but a direct comparison of the surrogate's predictions versus actual attention behavior would make the theoretical contribution much more credible.

- The unification in Appendix B is somewhat loose. The connections between existing methods and the capacity objective require stacking multiple simplifying assumptions (isotropic queries, identity noise, ignoring values, etc.) on top of the already-simplified linear-Gaussian model. The paper is upfront about this ("should be interpreted qualitatively"), but it weakens the claim that these methods are "approximations of the same objective" — they're more like "vaguely related under heavy simplification." A more honest framing would help.

- The improvements over EA (Expected Attention) are often marginal. Looking at Table 1 carefully, at compression ratio 0.25, CapKV beats EA by ~1-2 points on average for both Qwen3-8B and Qwen3-14B. The gap widens at higher compression, but at 0.25 and 0.5 the differences are small enough that they could be within noise on many individual tasks. The paper doesn't report confidence intervals or significance tests, which makes it hard to assess whether these gains are reliable.

---

> ### Author Rebuttal · Authors · 2026-03-31
>
> ## 1. Linear-Gaussian Model vs. Real Attention Model:
> We appreciate the reviewer’s comments on the linear-Gaussian approximation. While it simplifies the attention mechanism, we conducted empirical analyses to validate its utility for KV cache eviction tasks.
>
> ### Output Cosine Similarity
>
> To assess the approximation, we computed the cosine similarity between the outputs of the linear-Gaussian model ($Y_{CapKV}$) and the real attention mechanism ($Y_{full}$) on the Qwen3-8B model across different layers. The results showed that cosine similarities ranged from 0.6 to 0.9, indicating that the directionality of the two outputs is highly consistent. While magnitudes of $Y_{CapKV}$ and $Y_{full}$ differ due to the absence of Softmax normalization, the preservation of relevant directions is crucial for cache eviction.
>
> |Dataset|2wikimqa|qasper|trec|
> |-|-|-|-|
> |Layer2|0.60±0.0498|0.69±0.0380|0.71±0.0227|
> |Layer15|0.91±0.0247|0.91±0.0145|0.91±0.0121|
> |Layer20|0.86±0.0134|0.85±0.0098|0.88±0.0073|
> |Layer35|0.66±0.0486|0.74±0.0308|0.73±0.0298|
>
> ### Signal and Noise Independence Validation
>
> We also evaluated the independence of the noise term $\epsilon$ from the query vector $q$ by measuring Pearson’s correlation coefficient ($\text{Corr}(\epsilon, q)$) across layers and datasets. Correlations were close to zero (below 0.012), supporting our assumption that $\epsilon$ does not contain significant linear information about the query.
>
> |Layer|2wikimqa|qasper|trec|
> |-|-|-|-|
> |Layer2|0.0027±0.0035|0.0029±0.0035|0.0014±0.0033|
> |Layer15|0.0117±0.0031|0.0081±0.0029|0.0115±0.0024|
> |Layer20|\-0.0026±0.0035|\-0.0021±0.0030|0.0056±0.0025|
> |Layer35|0.0003±0.0064|0.0049±0.0060|0.0017±0.0032|
>
>
> ### Theoretical Validity and Maximum Entropy Principle
> From a theoretical standpoint, it is well known that the Gaussian distribution is the most robust and minimally restrictive model when only the first- and second-order moments (mean and covariance) of noise are known, according to the maximum entropy principle. Even when the actual noise distribution may exhibit non-Gaussian features (such as heavy tails), the log-determinant metric derived from this Gaussian assumption remains the optimal capacity measure for linear prediction scenarios. This is corroborated by the high Spearman correlation ($\rho \approx 0.8$) between our capacity predictions and downstream performance, which further validates the effectiveness of our information-theoretic framework.
>
> ## 2. Simplifying Assumptions:
> We appreciate the reviewer's observation on the simplifications made in our model. We fully acknowledge your point that existing methods can only be mapped to the CapKV framework under strong assumptions. However, we would like to emphasize that our primary objective is to provide valuable insights for guiding future method design. As such, we will revise our presentation in the final version to more clearly convey that these assumptions.
>
> ## 3. Addressing Data Fluctuations and Variance:
> We have included additional variance data for CapKV's performance across different tasks and compression rates:
>
> |CR|2WQA|GR|HpQA|Lcc|MN|MF|Mque|NQA|PC|PR|Qser|QMS|RBP|SSum|TREC|TQA|Avg.|
> |-|-|-|-|-|-|-|-|-|-|-|-|-|-|-|-|-|-|
> |0.5|44.49±0.066|33.04±0.033|61.40±0.179|59.67±0.278|24.51±0.057|51.05±0.042|33.96±0.085|28.90±0.377|10.67±0.236|98.04±0.269|41.95±0.075|24.22±0.019|63.47±0.108|39.71±0.047|66.50±0.000|87.27±0.156|48.05±0.067|
> |0.75|38.99±0.391|31.03±0.028|53.84±0.344|51.65±0.193|23.18±0.042|44.53±0.264|30.07±0.335|28.33±0.207|8.83±0.471|97.37±0.292|37.68±0.005|23.57±0.005|63.83±0.014|38.13±0.113|60.50±0.000|86.05±0.269|44.85±0.023|
> |0.9|31.42±0.047|27.19±0.057|49.56±1.325|39.58±0.047|20.22±0.033|35.23±0.627|25.05±0.047|24.55±0.033|6.83±0.236|56.83±0.387|27.54±0.033|21.52±0.061|62.79±0.193|35.31±0.217|36.00±0.000|85.25±0.509|36.56±0.007|
>
> These results indicate that CapKV's performance is stable across tasks, with minimal fluctuation, and we will include further details in the appendix.
>
> ## 4. Using Variance vs. Mean for Evaluation:
> Regarding your question about using the Conv variance instead of the mean, we conducted experiments comparing the use of the historical query mean and Conv variance in the scoring process. We found that using the mean generally resulted in more stable and superior performance. However, we acknowledge that variance might provide a closer approximation to our theoretical framework, though it tends to be more sensitive and less stable across tasks.
>
> **Performance Comparison**:
> |CR|Mean|Conv|
> |-|-|-|
> |0.25|49.51|49.23|
> |0.5|48.15|47.01|
> |0.75|44.88|42.28|
> |0.9|36.57|32.65|
>
> The results indicate that the use of the query mean provides more stable performance, while variance-based methods are more sensitive to task-specific fluctuations.
>
> We hope the response above addresses your concerns and kindly ask you to raise your score. Thank you again for helping us improve our work.

---

> > ### Author Rebuttal · Reviewer_NaPt · 2026-04-01
> >
> > Thank the authors for the rebuttal. I'll remain the positive score.

---

> > > ### Author Response · Authors · 2026-04-04
> > >
> > > We sincerely thank the reviewer for the positive evaluation and for recognizing our efforts in the rebuttal. We are glad that our responses have addressed the concerns. We greatly appreciate the reviewer’s support and constructive feedback, which will help us further improve the paper in the final version.

---

### Decision · Program_Chairs · 2026-04-30

**Decision:**

Accept (regular)

**Comment:**

This submission proposes an information-theoretic perspective on KV cache eviction and introduces CapKV as a practical eviction method. The reviewers recognized the novelty of the unifying perspective and the empirical performance of the proposed method across multiple models and benchmarks, while also raising concerns about the simplifying assumptions underlying the theoretical analysis. The information-theoretic framing is intriguing, but the reviewers questioned whether it constitutes a sufficiently rigorous foundation for the paper’s main claims. Although some questions remain regarding the exact correspondence between the theoretical surrogate and standard attention, the authors’ response addressed several experimental concerns. Overall, I view this as a worthwhile conceptual and practical contribution, and I therefore recommend acceptance.